# A low-cost and shielding-free ultra-low-field brain MRI scanner

Yilong Liu [1,2,7], Alex T. L. Leong[1,2,7], Yujiao Zhao[1,2,7], Linfang Xiao[1,2,7], Henry K. F. Mak [3], Anderson Chun On Tsang [4], Gary K. K. Lau[5], Gilberto K. K. Leung[4] & Ed X. Wu [1,2,6 ✉]

Magnetic resonance imaging is a key diagnostic tool in modern healthcare, yet it can be cost-prohibitive given the high installation, maintenance and operation costs of the machinery. There are approximately seven scanners per million inhabitants and over 90% are concentrated in high-income countries. We describe an ultra-low-field brain MRI scanner that operates using a standard AC power outlet and is low cost to build. Using a permanent 0.055 Tesla Samarium-cobalt magnet and deep learning for cancellation of electromagnetic interference, it requires neither magnetic nor radiofrequency shielding cages. The scanner is compact, mobile, and acoustically quiet during scanning. We implement four standard clinical neuroimaging protocols (T1- and T2-weighted, fluid-attenuated inversion recovery like, and diffusion-weighted imaging) on this system, and demonstrate preliminary feasibility in diagnosing brain tumor and stroke. Such technology has the potential to meet clinical needs at point of care or in low and middle income countries.

[1] Laboratory of Biomedical Imaging and Signal Processing, The University of Hong Kong, Pokfulam, Hong Kong SAR, China. [2] Department of Electrical and Electronic Engineering, The University of Hong Kong, Pokfulam, Hong Kong SAR, China. [3] Department of Diagnostic Radiology, Li Ka Shing Faculty of Medicine, The University of Hong Kong, Pokfulam, Hong Kong SAR, China. [4] Department of Surgery, Li Ka Shing Faculty of Medicine, The University of Hong Kong, Pokfulam, Hong Kong SAR, China. [5] Division of Neurology, Department of Medicine, Li Ka Shing Faculty of Medicine, The University of Hong Kong, Pokfulam, Hong Kong SAR, China. [6] School of Biomedical Sciences, Li Ka Shing Faculty of Medicine, The University of Hong Kong, Pokfulam, Hong Kong SAR, China. [7] These authors contributed equally: Yilong Liu, Alex T. L. Leong, Yujiao Zhao, Linfang Xiao. ✉email: ewu@eee.hku.hk

Magnetic resonance imaging (MRI) is widely considered as the most important medical imaging technology innovation in modern healthcare[1]. MRI is intrinsically superior to other imaging modalities, because it is non-invasive, non-ionizing, inherently quantitative and multi-parametric. As the human body is comprised of ~70% water, there is an abundance of protons that can be excited, manipulated and imaged by MRI. This enables physicians to visualize various types of tissues and assess their structural and physiological integrity. Over 150 million investigations with MRI are performed each year worldwide[2]. Examples of routine clinical MRI applications, include the diagnosis and prognosis of diseases (e.g., tumors, ischemic stroke and hemorrhage) and injuries in bodily systems (e.g., nervous, hepatobiliary, pancreatic, and musculoskeletal systems)[3]. The success of MRI utilization has been driven by the synergistic efforts by the clinicians, physicists, and engineers worldwide in their pursuit of quality and imaging capabilities[4–7]. Notable technical advances include superconducting magnet designs for small fringe field, low or no helium boil-off rate, and large bore size for patient comfort; development of powerful gradient and RF electronics to exploit the increased signal-to-noise ratio (SNR) at high field for speed and new contrasts; parallel signal receiving for fast imaging; and ultra-high-field MRI (7 T and higher) for scientific exploration and clinical applications[4,7].

However, MRI accessibility is low and extremely inhomogeneous around the world. According to the 2020 Organisation for Economic Co-operation and Development (OECD) statistics[8], there are approximately 65,000 installations of MRI scanners worldwide (~7 per million inhabitants) compared to ~200,000 for CT and ~1,500,000 for ultrasound scanners. The distribution of MRI scanners is concentrated mainly within high income countries with scarce availability in low and middle income countries. Hence, ~70% of the world's population have little to no access to MRI. This disparity highlights the cost-prohibitive nature of high-field superconducting MRI scanners (1.5 T and 3 T). First, these scanners rely on complex superconducting electromagnet/cryogenics designs and ever increasingly powerful electronics (including gradient and radiofrequency power systems) for fast imaging and/or advanced imaging features like brain functional MRI and diffusion tractography, yet routine clinical uses only necessitate a small portion of these imaging protocols[9,10]. Second, they require expensive installation due to infrastructural requirements (e.g., site preparation to host the large magnets that typically weigh 3000-4500 kg, magnetic shielding and radiofrequency shielding, emergency helium exhaust conduit, electricity to drive power-consuming electronics, and water requirement for gradient cooling). Third, they require a high maintenance cost for helium refill/re-liquification (a rare and dwindling non-renewable resource) and regular cold-head services. Last, these complex scanners require high operation costs for specialized radiographic technicians. Thus, the vast majority of clinical MRI scanners are placed in highly specialized radiology departments, large centralized imaging centers, and housed on ground floors of hospitals and clinics. This reality excludes easy access for neurology clinics, trauma centers, surgical suites, neonatal/pediatric centers, and community clinics. Ultimately, these factors present a major roadblock in MRI accessibility in healthcare.

Recently, there has been an impetus to develop low-cost MRI technologies at ultra-low-field (ULF) strengths[11–13], i.e., <0.1 T, for truly point-of-care applications. They include using resistive electromagnets that can produce a homogenous field (0.0065 T[14] and 0.023 T[15]), rotating Halbach permanent magnet array to produce an inhomogeneous field (0.077 T[16]), Halbach single-sided permanent magnet array at 0.064 T with inhomogeneous field[17], and magnet-free earth-field MRI imaging[18]. However, these designs have not demonstrated sufficient image quality or adequate versatility for clinical applications.

Despite its inherent limitations, we and others argue that ULF still holds clear potential in creating a new class of low-cost MRI technologies for accessible healthcare with scanners that are simple to onboard, maintain and operate[11–13]. In the past two years, intensive ULF MRI developments using permanent magnets (homogenous Halbach 0.05 T[19,20], double-pole 0.05 T[21], and rotating inhomogeneous Halbach 0.08 T[22]) are promising. Their results suggest the possibility of generating brain images with low-cost hardware, though the imaging versatility and image quality remain unknown. ULF MRI developments eliminate the need for a magnetic shielding cage because of dramatic fringe field reduction, yet they all require the bulky radiofrequency (RF) shielding cage in practice to prevent external electromagnetic interference (EMI) signals during ULF MRI scanning. Several solutions have been recently proposed to tackle the EMI problem for ULF MRI without RF shielding room. One group used magnetometers to sense environmental EMI and remove EMI signal in MRI receive coil via an adaptive suppression procedure[23]. The method was hardware demanding and only yielded limited success. Another study utilized simple conductive cloth to cover the subject during scanning[19]. This passive method could alter and reduce EMI from external environments, but its performance was suboptimal. An analytical approach was proposed to estimate EMI signal in MRI receive coil from EMI signals detected by EMI sensing RF coils based on the frequency domain transfer functions among coils[24]. More recently, it was extended for time domain implementation as linear convolutions and with an adaptive procedure[25]. The method eliminated EMI substantially but could only produce very satisfactory brain imaging results when used together with conductive cloth and body surface electrode for EMI pickup. We also note the recent commercial endeavor by Hyperfine (www.hyperfine.io/portable-MRI). Its FDA-approved portable 0.064 T brain MRI scanner can operate in unshielded environments using a proprietary EMI removal method. The scanner has shown potential for point-of-care applications especially in the intensive care unit (ICU) and COVID-19 wards[26], which has clearly illustrated the unmet needs within vital emergency healthcare where patients require easy and rapid access to MRI.

In this study, we report the development and initial clinical demonstration of a permanent magnet-based, low-cost, low-power, and shielding-free brain ULF MRI scanner. Specifically, we first designed our system around the framework of a homogeneous 0.055 T permanent double-pole magnet and linear imaging gradients. This configuration allows us to form images with various universally adopted contrasts and flexible orientations for clinical brain imaging, including fluid-attenuated inversion recovery (FLAIR) like and diffusion-weighted imaging (DWI). It also enables a high level of flexibility in developing future ULF MRI protocols by building upon the methodologies developed over the past three decades for high-field MRI scanners. Second, we developed a deep learning driven EMI cancellation technique to model, predict and robustly remove the external and internal EMI signals from MRI signals, thus eliminating the traditional RF shielding cage. Third, we succeeded in implementing the four essential protocols on this low-cost prototype scanner for clinical brain MRI, namely, T1-weighted (T1W), T2-weighted (T2W), FLAIR-like, and DWI with isotropic diffusion weighting. Last, we demonstrated the preliminary feasibility in diagnosing tumor and stroke cases as compared to 3 T clinical MRI results.

**Fig. 1 Prototype of a low-cost low-power and shielding-free brain ultra-low-field (ULF) magnetic resonance imaging (MRI) scanner with homogenous 0.055 Tesla magnetic field and small 5 Gauss fringe field. a** With ~2 m² footprint and no radiofrequency/magnetic shielding cages, the scanner can be mobile and located on any building floor with no special siting requirements. It operates from only a standard alternating current (AC) wall power outlet (50 Hz 2-phase 220 V 15 A). **b** The 0.055 T magnet utilizes an iron yoke, samarium-cobalt (SmCo) plates, polar pieces, anti-eddy current plates and passive shimming rings, and with adequate opening for patient chest and shoulder (29 cm vertical gap and 70 cm width). **c** The magnet provides a homogeneity <2000 ppm peak-to-peak over 240 mm diameter of spherical volume (DSV), which can be reduced to <250 ppm after additional passive shimming. The 5 Gauss fringe field is within 45 cm, 90 cm, and 80 cm in X, Y and Z directions from magnet center.

## Results

**0.055 T brain ULF MRI system hardware design**. We demonstrated the technical feasibility of a new class of cost-effective MRI technology by designing and constructing a prototype adult brain ULF MRI scanner that operates on a standard alternating current (AC) wall power outlet (two-phase 220V 15A) in absence of any RF and magnetic shielding cages. The system was based on a compact two-pole 0.055 T permanent samarium-cobalt (SmCo) magnet with front opening of 29 cm height and 70 cm width for patient chest and shoulder access (Fig. 1a; see methods section for more detailed descriptions and specifications of magnet design, and gradient and RF subsystems). The footprint of the main components (i.e., magnet and electronic cabinet) occupied an area of ~2 m². The system required no magnetic shielding. Key magnet components (yoke, SmCo plate, pole, anti-eddy current plate, and shimming ring; Fig. 1b) were designed to provide a 0.055 T field with inhomogeneity <2000 ppm peak-to-peak over 240 mm diameter of spherical volume (DSV) (Fig. 1c), which was subsequently reduced to <250 ppm after additional passive shimming. The 5 Gauss fringe field was within 45 cm, 90 cm, and 80 cm in X, Y and Z directions from magnet center (Fig. 1a). The magnet assembly weighted about 750 kg. Here, we employed standard low-cost and off-shelf electronics for simplicity, including MRI console (www.mrsolutions.com) and gradient amplifier (https://pcipa.com).

Although the cylindrical Halbach magnet offers a lighter weight and smaller fringe field[19,20,22], we chose the open double-

pole magnet design for patient comfort. These Halbach magnet designs subject patient head to a tight and enclosed space inside magnet and restrict access to patient during scanning. Further, we chose SmCo instead of neodymium-iron-boron (NdFeB), the most commonly used rare earth material for permanent MRI magnets[11]. Despite the relatively lower maximum energy product (BH$_{max}$), SmCo offers significantly improved temperature stability arising from its extremely low temperature remanence coefficient (SmCo 0.015%/ºC vs. NdFeB 0.125%/ºC[27,28]), which eliminated the need for any magnet temperature regulation schemes to stabilize temperature-dependent field.

Our engineered system requires no RF shielding cage. To tackle the EMI signals from the external environments and internal low-cost electronics during scanning, we developed a deep learning driven EMI cancellation scheme (Fig. 2). In unshielded environments, EMI signal can change dynamically during scanning due to surrounding EMI sources of various nature and behaviors. Further, EMI signal received by MRI receive coil can be influenced by the human body, which serves as an effective antenna[29,30] for EMI pickup. Body position change during scanning can also affect EMI signal. Therefore, a data driven method such as deep learning was preferred over an analytical approach (such as the recently emerged ones[24,25]) to provide a more robust, yet relatively simple, EMI prediction and removal.

Ten EMI sensing coils, strategically placed around scanner and inside electronic cabinet, simultaneously acquired radiative EMI signals only in the absence of any RF shielding for the scanner

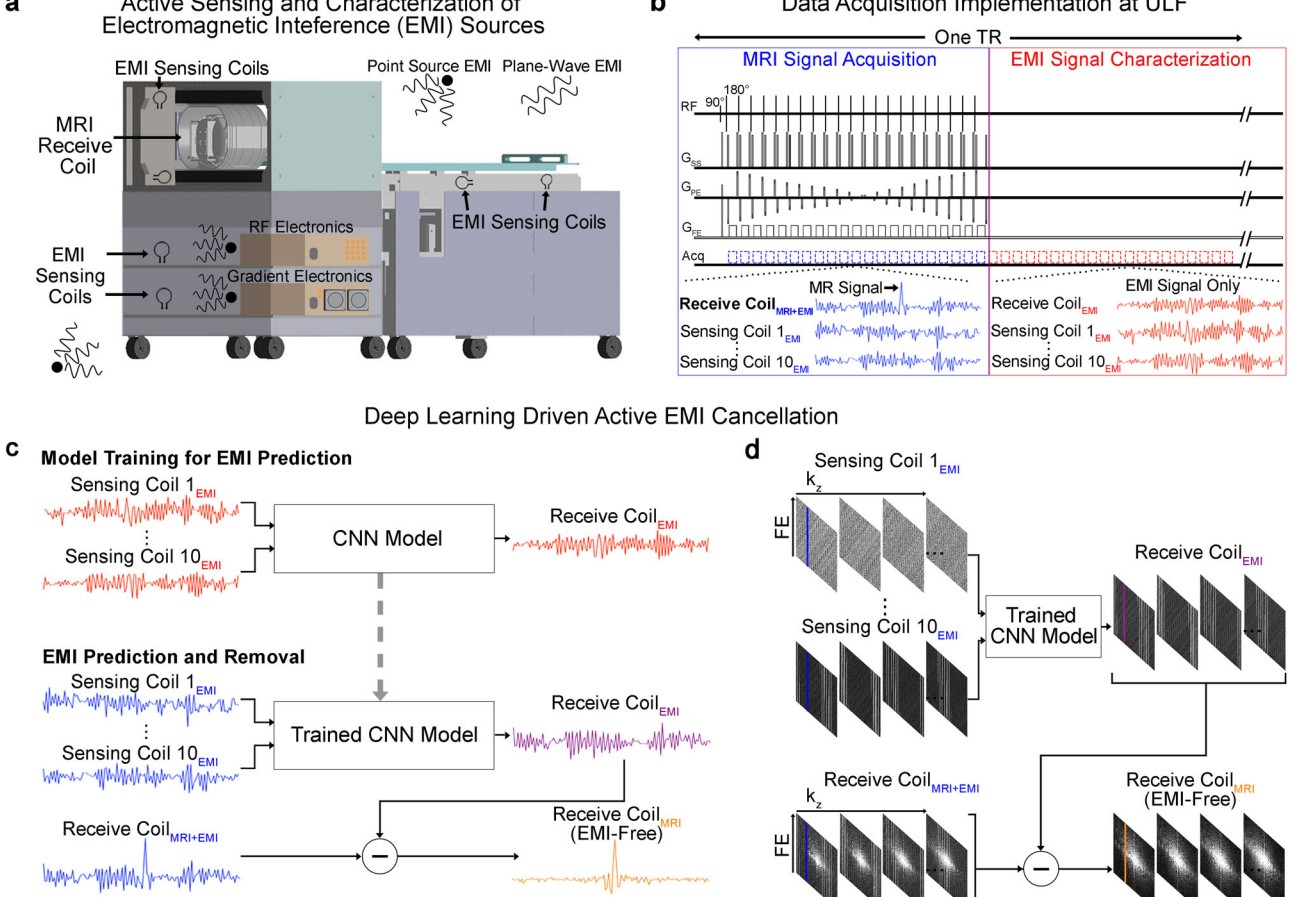

**Fig. 2 Deep learning driven detection and cancellation of electromagnetic interference (EMI) signals for low-cost 0.055 T MRI without radiofrequency (RF) shielding. a** Ten small RF coils, i.e., EMI sensing coils, are strategically placed in the vicinity of transmit and receive RF coils, underneath the patient bed, and inside electronic cabinet to actively detect EMI signals only that are from both external environments and internal electronics. **b** Illustration of the 3D fast spin echo (FSE) pulse sequence data acquisition for both MRI signal collection and EMI signal characterization. Within each time of repetition (TR), data are acquired simultaneously by both MRI receive coil and EMI sensing coils across within two windows—the conventional MRI signal acquisition window and the EMI signal characterization window. Note that MRI signal is zero within EMI signal characterization window. **c** After each scan, a convolutional neural network (CNN) model is trained to establish the relationship between the EMI signals received by MRI receive coil and the sensing coils within EMI signal characterization window. The trained model can then predict the EMI signal component detected by the MRI receive coil within the MRI signal acquisition window from the simultaneously detected signals by EMI sensing coils for each frequency encoding (FE) line. The predicted EMI is subtracted from MRI receive coil signal to produce the EMI-free FE line. **d** This procedure is repeated for all individual FE lines before averaging and subsequent image reconstruction from the EMI-free k-space data.

(Fig. 2a, b). Signals detected by these EMI sensing coils and main MRI receive coil within EMI signal characterization window retrospectively trained a convolutional neural network (CNN) model. This model could then predict the EMI signal component in MRI receive coil signal for each frequency encoding (FE) line within MRI signal acquisition window based on the EMI signals simultaneously detected by EMI sensing coils. This predicted EMI signal component was subsequently subtracted or removed from the MRI receive coil signals, creating EMI-free k-space data prior to image reconstruction (Fig. 2c, d). With this deep learning EMI cancellation procedure, we eliminated the undesirable EMI signals in a highly reliable and robust manner for both phantom and human brain imaging (Fig. 3), even when environmental EMI sources and their spectral characteristics changed dynamically during scanning (Supplementary Figs. 1 and 2) The performance of this deep learning EMI cancellation procedure was also experimentally compared to the ground truth scenario where a RF shielding cage was employed to fully enclose the subject during scanning. The results in Supplementary Fig. 3 demonstrated that, in absence of RF shielding cage, the procedure

provided a nearly complete removal of EMI noise in the images, with final image noise levels as low as those obtained using the RF shielding cage (within 5% range).

**Imaging protocol implementation using 0.055 T brain ULF MRI.** High-field superconducting MRI research in the past four decades has led to numerous contrasts and protocols to probe brain structures, physiology and functions at different levels. Yet, the most valuable and universally adopted protocols in clinical brain imaging workflows are T1W, T2W, FLAIR, and DWI[31–34], which presently account for the majority of clinical brain MRI scans[9]. We focused on implementing and optimizing these four critical clinical protocols. T1W protocol used a 3D gradient-echo (GRE) sequence with TR/TE = 52/13 ms, while T2W and FLAIR protocols used a 3D fast spin echo (FSE) sequence with TR/TE = 1500/202 ms and 500/129 ms, respectively. Further, DWI protocol with 3D isotropic diffusion weighting was successfully implemented with TR/TE = 2800/102 ms and isotropic diffusion weighting factor b-value 500 s/mm$^2$ using a 2D echo planar imaging (EPI) sequence. DWI is a technically challenging but

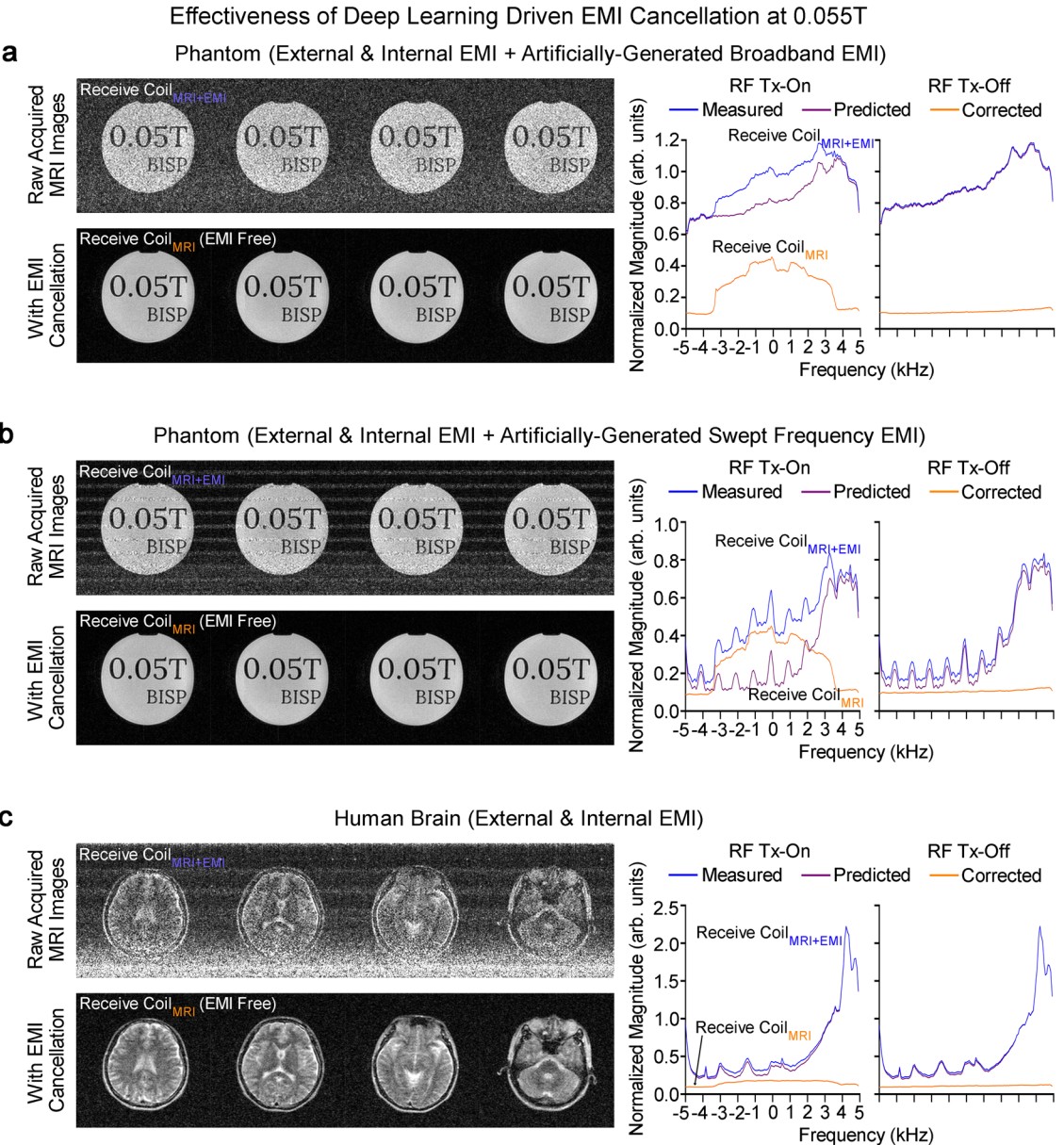

**Fig. 3 Robust deep learning driven EMI cancellation for 0.055 T phantom and brain imaging without RF shielding in the presence of various external or internal EMI sources.** Representative 3D FSE T2-weighted (T2W) images (left) and corresponding magnitude averaged spectra (right) of raw MRI FE lines, before and after EMI elimination, with RF transmit (Tx) power on (i.e., producing MRI signals) or off (i.e., no MRI signal, EMI signals only) for **a** phantom with a broadband EMI generated from a nearby source, **b** phantom with a swept frequency EMI generated from a nearby source, and **c** human brain (23 yrs. old; male). The proposed deep learning EMI elimination technique can robustly predict the EMI signals detected by the MRI receive coil, enabling MRI scan without any RF cage or shield. Note that, during brain scanning, the large human body acts as an antenna that receives high level of external environmental EMI signals, which can still be effectively eliminated by the technique.

clinically valuable protocol particularly for early stroke diagnosis. Careful hardware calibration and compensation procedures (e.g., gradient eddy current correction) were performed on our 0.055 T MRI scanner, when implementing FSE and EPI sequences. FSE and EPI are the most common sequences in high-field clinical MRI. We optimized the four scan protocols for both image SNR and contrast characteristics similar to those of clinical high-field MRI.

Typical images from healthy adult subjects are shown in Figs. 4 and 5. Four-contrast image datasets were acquired within a total scan time of 30 mins (5.5 mins, 7.5 mins, 7.5 mins, and 9.5 mins for T1W, T2W, FLAIR, and DWI, respectively). Acquisition image resolutions were ~2 × 2 × 10 mm³ and ~4 × 4 × 10 mm³

for T1W/T2W/FLAIR and DWI, respectively. Reconstructed image resolution was 1 × 1 × 5 mm³ for all protocols for better visualization effect. The AC power consumption during scanning was low (<1200 W). Scanning was acoustically quiet at 0.055 T with maximum peak sound pressure level (SPL) <85 dBA (Supplementary Fig. 4) while it was reported previously that the acoustic noise could reach up to 120 dBA at 3 T[35]. Note that, compared with high-field images, less gray matter and white matter contrast was observed in these T1W, FLAIR and DWI images, likely because of decreased gray and white matter contrast, increased noise level and strong partial volume effect associated with large voxel size at 0.055 T. This lack of apparent gray and white matter contrast was also ascribed partly to our

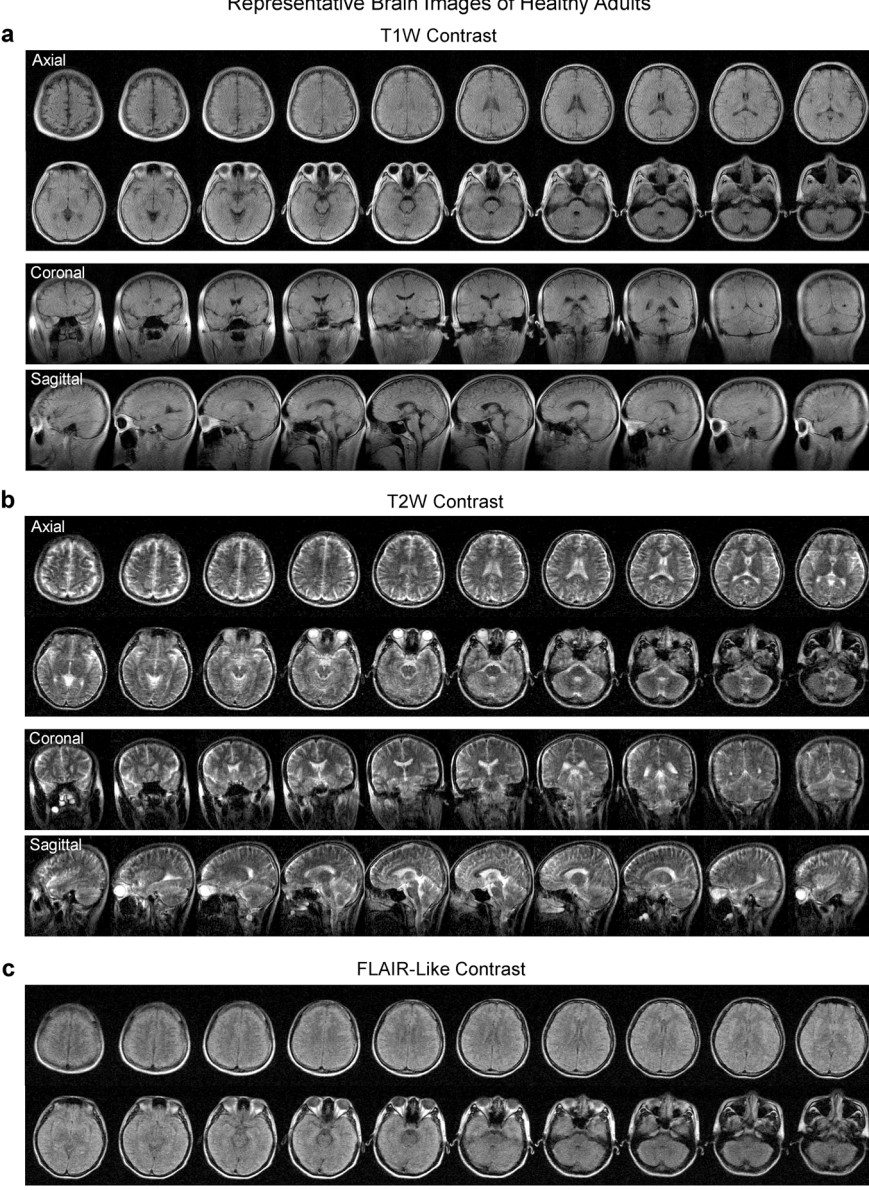

**Fig. 4 Typical brain images from healthy adults produced by the low-cost and shielding-free 0.055 T MRI scanner.** Whole-brain axial, coronal and sagittal sections of the brain were acquired at two common contrasts, namely **a** T1-weighted (T1W) images using the 3D gradient-echo sequence and **b** T2W using 3D fast spin echo sequence. **c** Whole-brain axial sections acquired with fluid-attenuated inversion recovery (FLAIR) like contrast using the 3D FSE sequence with short TR. Scan times are 5.5 mins, 7.5 mins and 7.5 mins for T1W, T2W and FLAIR protocols, respectively. All images are displayed at a spatial resolution of 1 × 1 × 5 mm$^3$, while the acquisition resolution is approximately 2 × 2 × 10 mm$^3$. The axial (23 yrs. old; male) and coronal/sagittal (23 yrs. old; male) images shown here were acquired from two healthy adults, respectively.

prioritization for image SNR during protocol optimization. In addition to the four imaging protocols, we also demonstrated the potential of acquiring images using other complex and hardware-demanding sequences such as true fast imaging with steady-state free precession (TrueFISP) (Supplementary Fig. 5), showing the potential of implementing more advanced MRI methods, such as TrueFISP based MR fingerprinting[36].

**Demonstration of clinical utility in tumor and stroke patients.** T1W and T2W images present the classical contrasts for anatomical structures and abnormal tissue differentiation. FLAIR attenuates cerebral spinal fluid (CSF) signals, while the remaining T2-weighted signals are highly effective in delineating brain pathologies like cortical, periventricular, and meningeal diseases.

DWI probes the tissue microstructures through water molecular diffusion and is extremely sensitive to early ischemic stroke, so it remains the most specific clinical protocol and gold standard in stroke imaging[32,34,37]. We evaluated the preliminary feasibility for the 0.055 T ULF MRI scanner to diagnose several major neurological diseases in 25 patients, including 13 with brain tumors, 8 with ischemic stroke, and 4 with intracerebral hemorrhage (ICH). For comparison, we also scanned these patients using a clinical 3 T MRI scanner. One senior clinical radiologist (H.K. Mak) read and reported the 0.055 T image findings in a blind manner.

We found that 0.055 T ULF images produced findings highly like those obtained with 3 T MRI. Our 0.055 T ULF MRI scanner detected most key pathologies (i.e., the characteristics of signal changes at pathological regions across different contrasts) in 25 patients studied. For example, in a brain tumor case, both 0.055 T

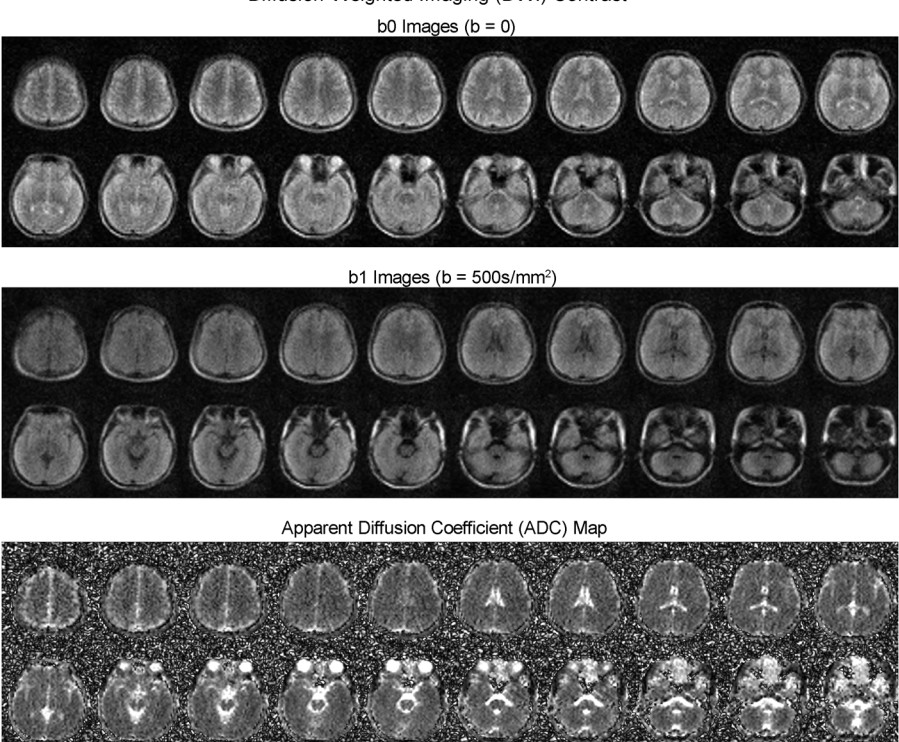

**Fig. 5 Typical diffusion-weighted imaging (DWI) images (from a healthy adult) produced by the low-cost and shielding-free 0.055 T MRI scanner.** Whole-brain axial sections (23 yrs. old; male) using the 2D echo-planar imaging DWI sequence with isotropic diffusion weighting factor b-value = 0 (b0 images) and 500 s/mm$^2$ (b1 images). Apparent diffusion coefficient (ADC) maps are also shown. Total scan time is 9.5 mins. All images are displayed at a spatial resolution of 1 × 1 × 5 mm$^3$ while the acquisition resolution is approximately 4 × 4 × 10 mm$^3$.

and 3 T images showed a mass in the right parietal cortex that was hypointense in T1W and hyperintense in T2W (Fig. 6a). The mass was extra-axial in nature (i.e., meningioma) based on the 3 T images. Meanwhile, 0.055 T FLAIR images showed weak but visible tumor contrast from gray matter while 0.055 T apparent diffusion coefficient (ADC) images corresponded well with 3 T ADC images. In addition to meningioma, the most common brain tumor found in clinics, we also demonstrated the ability to detect rare intraventricular cystic lesions. T1W images at both 0.055 T and 3 T showed clear asymmetry between the right and left hemisphere, especially in the right lateral ventricle, indicating the presence of a cystic lesion (Supplementary Fig. 6A). 0.055 T FLAIR contrast also exhibited excellent sensitivity and correspondence with 3 T results in further detecting a choroid plexus cyst (i.e., hyperintense signal) located at the occipital horn of the right lateral ventricle. We further showed in a separate patient the ability to detect subdural effusion due to past head trauma at bilateral fronto-parietal regions, showing as a bright signal in 0.055 T and 3 T T2W images (Fig. 6b). In the ischemic stroke case, the 0.055 T images were acquired 3 weeks after the 3 T MRI scan conducted two days after index stroke. Here, the ischemic infarct in the right parietal cortex was mildly hyperintense in 0.055 T DWI and hyperintense in 3 T DWI, while ADC showed slight hyperintensity at 0.055 T and hypointensity at 3 T (Fig. 7a). These results are highly consistent with the expected progression of ischemic stroke across different timepoints[32] (i.e., subacute vs. acute). Both T2 and FLAIR showed hyperintense signals at the infarct region. In addition to the large (>10 mm) ischemic infarcts, we could observe small lacunar infarcts (~3 mm) in a separate chronic stroke case in 0.055 T images (Supplementary Fig. 6B). In a subacute ICH case, a hematoma was identified at the left occipital lobe in both 0.055 T and 3 T images (Fig. 7b). The

hematoma exhibited a hyperintense rim with a hypointense core in T2W images, indicating blood product and/or hemosiderin of different stages at the rim and core regions. 0.055 T FLAIR and ADC images showed good correspondence with 3 T, though 0.055 T T1W images showed weaker contrast at the hematoma (i.e., hyperintense rim and hypointense core).

We also sought to demonstrate the advantage of brain imaging at 0.055 T with regard to metal implants. We imaged metallic clips and cerebrovascular stents that are commonly used in the brain for managing cerebral aneurysms and vascular stenosis, respectively (Fig. 8). Little susceptibility artifacts were observed in 0.055 T images, in contrast to the severe image artifacts expected at high field[38].

## Discussion

In this study, we developed and demonstrated a low cost, ultra-low-field 0.055 T MRI scanner that operated out of a standard AC wall power outlet only. Such scanner can be made low cost to manufacture, maintain and operate. For quantity production, we estimate hardware material costs under USD20K. It is compact, potentially mobile, and acoustically quiet during scanning. We designed an effective and simple EMI elimination method to enable MRI scanning without any RF shield or cage. We succeeded in implementing four essential and standard high-field MRI contrast protocols widely used for clinical brain imaging, and demonstrated their potential clinical utility.

**Promises of imaging at ultra-low-field.** The high cost of procuring, siting/installing, maintaining and operating the current clinical scanners constitutes a major roadblock in MRI accessibility in healthcare. Low-cost, low-power, compact, open, and

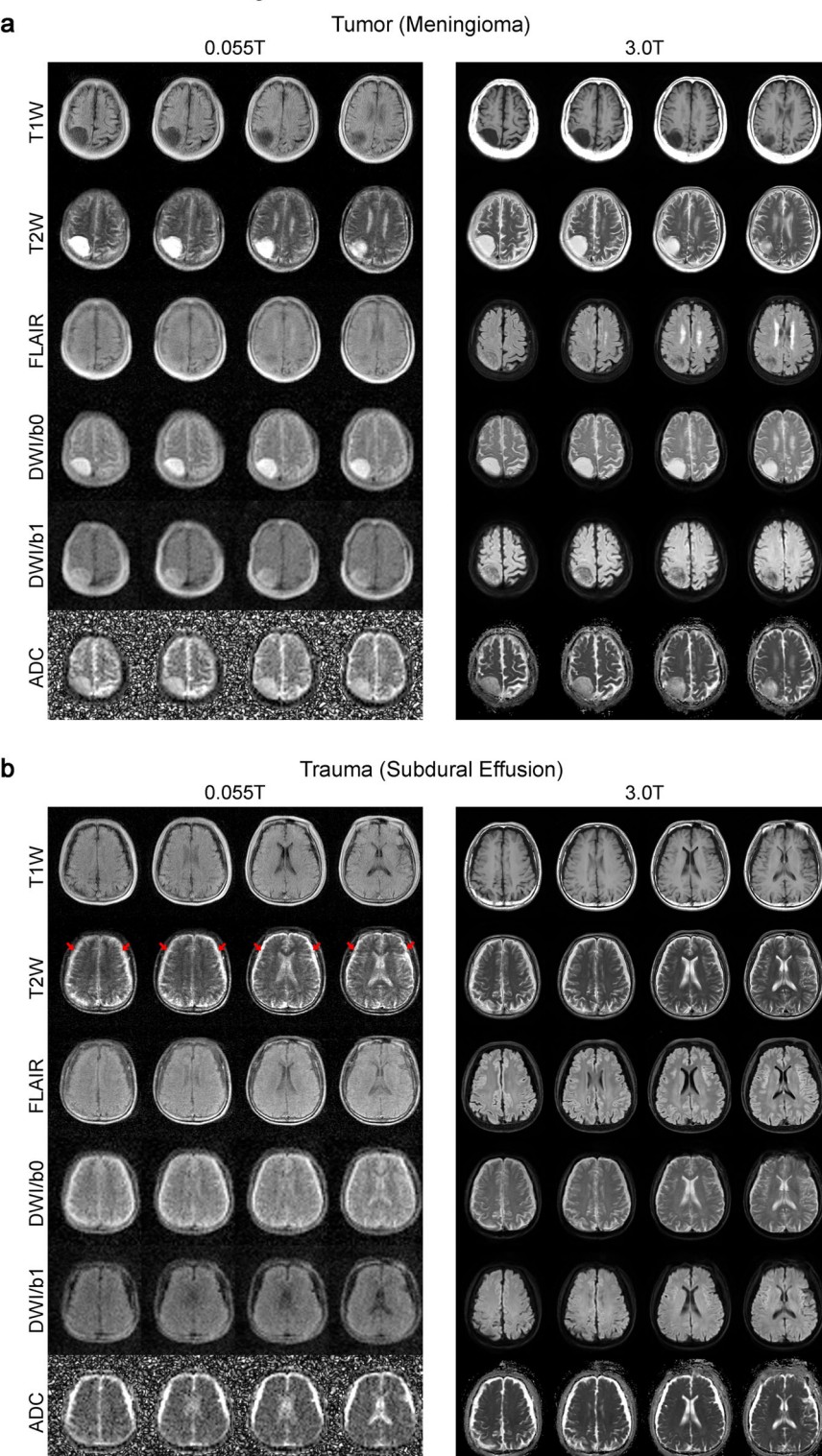

**Fig. 6 Clinical utility of 0.055 T MRI for examining tumor and trauma patients.** Total scan time was ~30 mins for T1W, T2W, FLAIR and DWI protocols for each patient at 0.055 T. Both patients were scanned by 3 T MRI on the same day using the standard T1W, T2W, FLAIR and DWI brain protocols (~20 mins total scan time) for comparison. **a** Patient (75 yrs. old; female) with an extra-axial mass (i.e., meningioma) at the right parietal cortex. Both 0.055 T and 3 T images showed that the tumor mass was hypointense in T1W and hyperintense in T2W. **b** Patient (64 yrs. old; male) with subdural effusion (i.e., collection of cerebrospinal fluid, CSF, trapped between the surface of the brain and the dura matter) due to previous head trauma. The subdural collection with sulci obliteration showing as a bright signal in T2W images (indicated by red arrows) was visible at bilateral fronto-parietal regions.

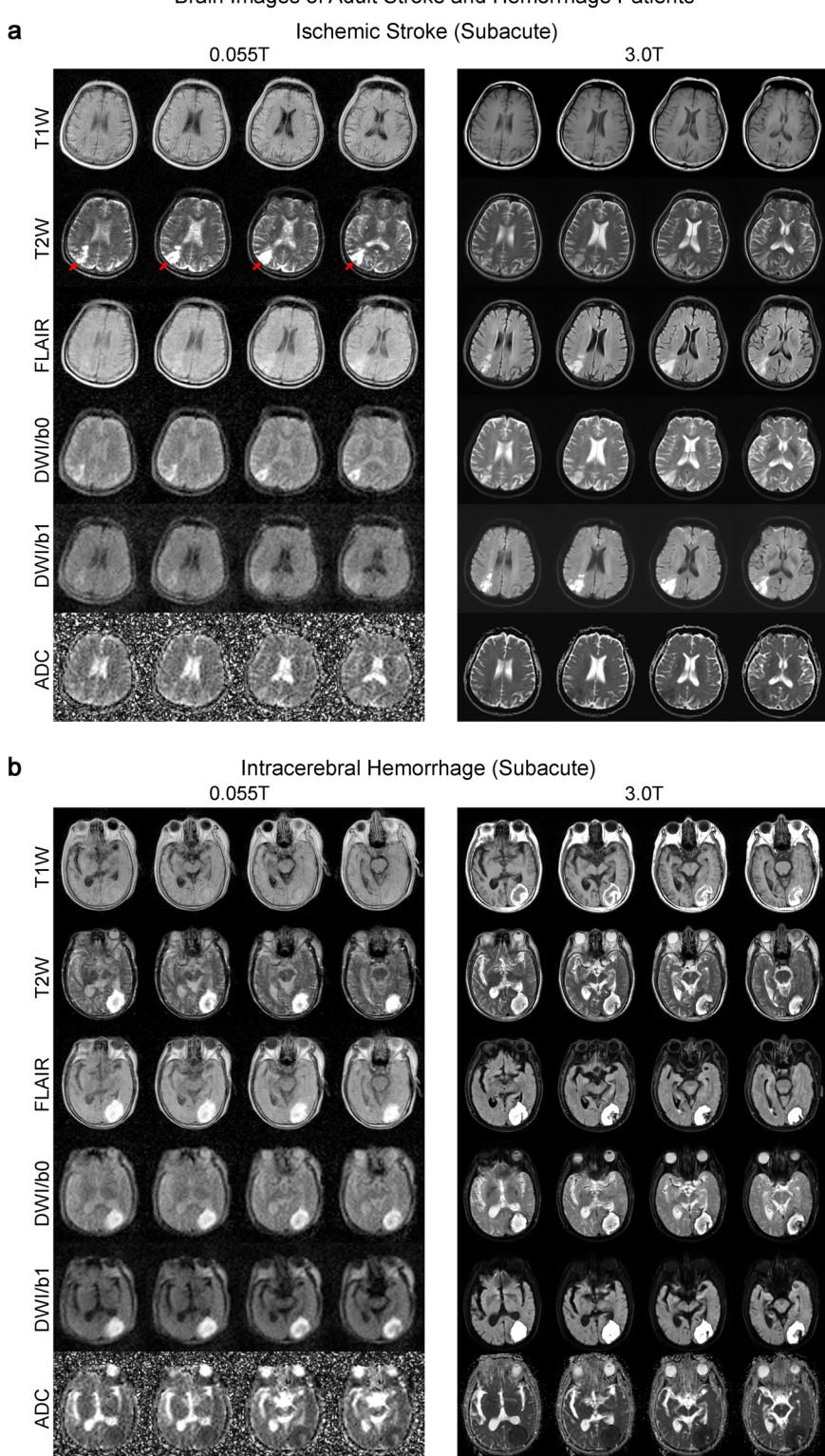

**Fig. 7 Clinical utility of 0.055 T MRI for examining ischemic stroke and intracerebral hemorrhage (ICH) patients.** Total 0.055 T scan time was ~30 mins for each patient. The patients were also scanned by a clinical 3 T MRI using the standard clinical protocols (~20 mins total scan time). **a** Subacute (~3 weeks) ischemic stroke patient (67 yrs. old; male). Note that 3 T clinical brain images were acquired 3 weeks prior to those acquired at 0.055 T. Ischemic infarct in the right parietal cortex (indicated by red arrows) was hyperintense in T2 and FLAIR images at 0.055 T and 3 T. Further, infarct was mildly hyperintense at ULF DWI, whereas it was hyperintense at 3 T. ADC showed slight hyperintensity at 0.055 T and hypointensity at 3 T. The corresponding signal changes for DWI and ADC maps at 0.055 T and 3 T are highly consistent with the expected progression of ischemic stroke across different timepoints (i.e., subacute vs. acute)[32]. **b** Subacute (~3 weeks) ICH patient (81 yrs. old; male). 0.055 T and 3 T images were acquired on the same day. Hematoma was visible in the left occipital lobe at 0.055 T and 3 T. It showed a hyperintense rim with a hypointense core in T2W images, indicating blood product and/or hemosiderin of different stages at the rim and core regions.

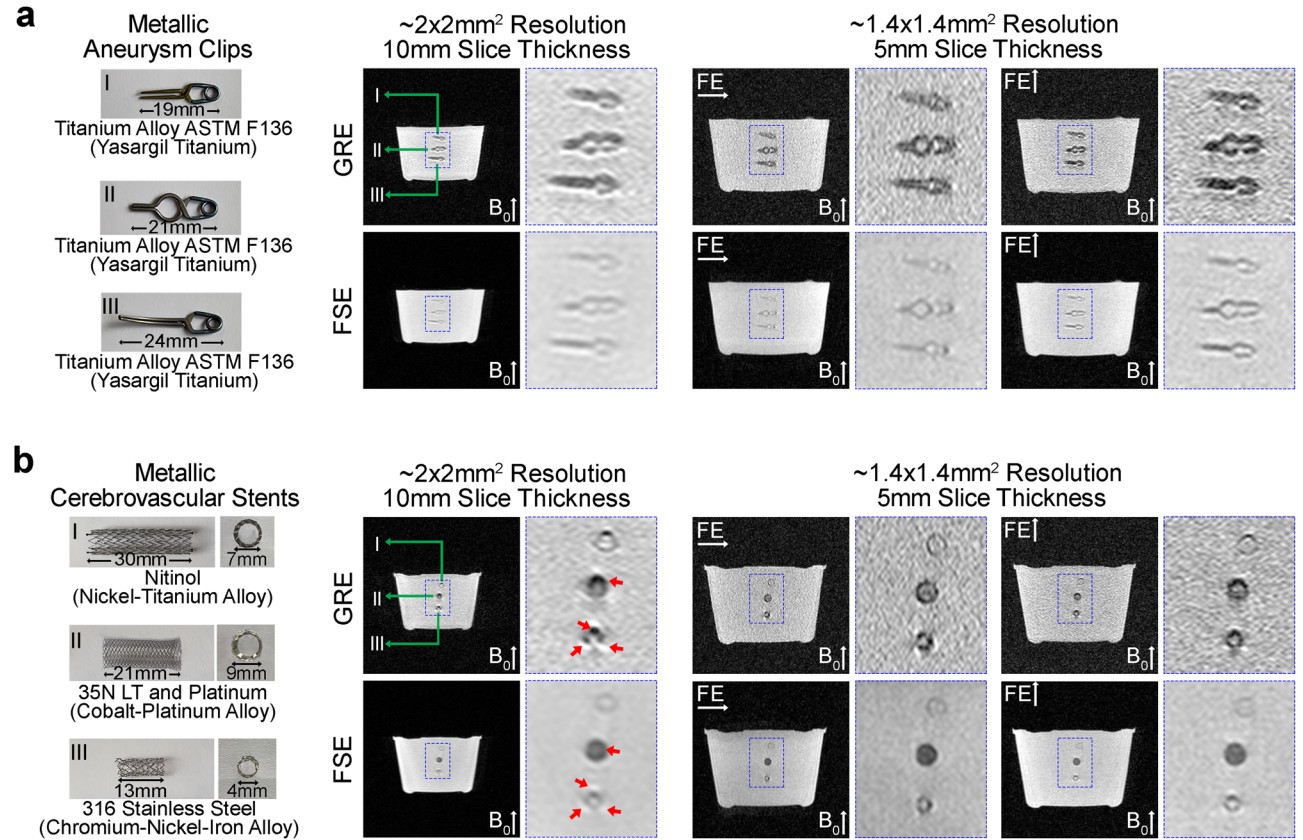

**Fig. 8 Low sensitivity to commonly used clinical metal implants at 0.055 T.** Illustration of metal implants (left) and corresponding images acquired at 0.055 T (right) of **a** titanium alloy aneurysm clips and, **b** cerebrovascular stents with three distinct types of metal alloys. Metal-induced image artifacts were dramatically reduced at 0.055 T. Almost no visible artifacts were present around the titanium alloy metal clips and the nickel-titanium alloy stent. Slight artifacts were visible in the images for the cobalt-platinum alloy and stainless-steel stents (indicated by red arrows). These implants were immersed in water and imaged with the 3D axial GRE and FSE sequences with FE direction along horizontal or vertical direction. Note that the main field 0.055 T was along the vertical direction.

shielding-free ULF MRI for brain imaging as demonstrated here aims to complement rather than compete with existing high-performance clinical MRI in healthcare.

ULF MRI holds several inherent attractions when compared to high-field MRI. They include open magnet configuration for patient comfort, low acoustic noise levels during scanning, low sensitivity to metallic implants, less image susceptibility artifacts at air/tissue interfaces, and extremely low RF specific absorption rate (SAR). Conventional tunnel-shaped superconducting high-field MRI scanners produces high acoustic noise levels (with maximum SPL up to 120 dBA at 3 T) during scanning because of high field strength and fast gradient switching[35,39]. Acoustic noise[35,39] and claustrophobia[40,41] remain a long-standing issue for patient comfort, accessibility, and safety. Up to 15% of patients who undergo MR examination in conventional scanners suffer from claustrophobia[40,41]. High acoustic noise levels during MRI can increase the risk of permanent hearing loss, cause temporary hearing threshold shift even with the standard MRI hearing protection measures[35,39], and induce anxiety[42,43], which often renders MRI unsuitable for several patient groups, such as those with sensory hypersensitivity disorders, neonates, and young children. Together with complete elimination of RF and magnetic shielding cages, our ULF MRI approach can offer a more patient-friendly alternative.

Metal implants or fragments cause undesired image artifacts and often constitute a major safety concern during MRI examination[44]. This presents a challenge to clinical MRI, because many patients with implanted devices are often those that require

frequent imaging evaluation and monitoring. Moreover, the prevalence of cerebrovascular, cardiac and/or orthopedic metal implants will continue to rise in an increasingly aging society[45], posing another barrier to future MRI accessibility. In using ULF (<0.1 T) and low-field (<0.5T), metal implants not only exhibit fewer artifacts, but also experience significantly less mechanical forces and RF-induced heating[46,47]. We and others showed that the presence of paramagnetic (i.e., titanium and titanium alloys) and ferromagnetic (i.e., cobalt, nickel and associated alloys) materials in aneurysm clips and cerebrovascular stents did not induce gross artifacts in the 0.05 T[46] and 0.055 T images (Fig. 8). We expect that ULF will enable MRI scanning of patients with medical metal implants or accident related metal fragments, who would otherwise not be candidates for conventional high-field MRI.

From biophysics perspectives, MRI at ULF presents several under-appreciated opportunities due to the unique tissue nuclear magnetic resonance (NMR) properties at such low field strengths. For example, tissues are expected to exhibit dramatically shorter T1[48,49] and longer proton T2 (as well as T2*)[50] at ULF. This leads to more time-efficient data acquisition protocols given the significantly faster longitudinal magnetization recovery (during repetitive excitation) and slower transverse magnetization decay (during signal readout with various encoding schemes). We have experimentally estimated in our preliminary study that the apparent T1/T2 values for gray matter and white matter were approximately 330/110 ms and 260/100 ms at 0.055 T (vs. 1300/110 ms and 830/80 ms at 3 T[51]) while CSF maintains long T1

(>1500 ms) and T2 (>1000 ms). Further, potential new contrasts remain to be discovered at ULF. For example, contrast-free imaging of tumors may be possible because of their altered T1 relaxation mechanism at ULF owing to water proton dynamics, e.g., the transmembrane exchange between the intracellular and extracellular compartments[52,53]. Relaxometry measurements indicated that proton T1 of tumor cells in an animal model was significantly longer at ULF (<0.1 T)[52]. It is also imperative to assess the use of existing clinical MRI contrast agents (i.e., gadolinium chelates) and explore the low-dosage applications of other exogenous contrast agents such as super-paramagnetic iron oxides (SPIOs) agents that exhibit high relaxivity effects at ULF[54].

**Challenges of imaging at ultra-low-field**. Key technical challenges in ULF MRI are the adequate image SNR, contrast, and resolution achievable within a reasonable scan time. At present, though partly alleviated by increased acquisition efficiency due to generally shorter T1 values, image SNR, contrast, and resolution at 0.055 T may be still too low for robust clinical applications. The ability of ULF MRI to differentiate various tissues and pathologies ultimately depends on the contrast to noise ratio and image resolution. The relative difference in T1 value between gray matter and white matter becomes smaller at ULF[55] when compared with that at high field[51]. This reduction of intrinsic gray and white matter contrast, together with the relatively low image SNR and resolution, can substantially limit the eventual gray and white matter contrast to noise ratios in ULF T1W images. Thus, it is imperative to tackle these fundamental diagnostic imaging issues in the future.

We envision various developments in hardware, image acquisition/reconstruction and sequence contrast schemes to tackle these issues. First, at ULF, the noise in MRI signals is dominated by the RF receiver coil noise, while the sample noise is negligible[56]. Therefore, ULF SNR can be significantly increased by cooling the RF receiver coil and the RF preamplifier, which can be potentially implemented by cryogenic cooling or cryogen-free conduction cooling using cryocoolers[57]. Second, the recent advances in deep learning image reconstruction from less or noisy k-space data[58,59] will greatly benefit future ULF MRI development. For example, we can exploit omnipresent mutual anatomical features across human brains. Conventional MRI data acquisition and image reconstruction procedures do not utilize any prior information on human anatomy. Thus, the genetically pre-defined and highly homogenized human brain anatomical information has been completely neglected in past MRI formation process, yet it can be potentially utilized to intelligently drive and drastically enhance MRI data acquisition, image reconstruction, and clinical utility to expedite MRI scan and tackle the low SNR challenge at ULF.

Meanwhile, a separate challenge for imaging at ULF is the clinical adoption and integration into various healthcare settings for specific applications in disease diagnosis, prognosis, or/and treatment monitoring. First, the best field strength for any ULF MRI should provide an optimal trade-off between scanner cost, weight, image quality, and specific clinical applications. It remains unresolved whether 0.055 T MRI is sufficient and optimal for clinical adoption. From our experience, we foresee that the rare earth material NdFeB can be incorporated into the magnet design in future development phase to increase field strength without increasing magnet weight and size, because NdFeB has much higher maximum energy product ($BH_{max}$) than SmCo. If so, the NdFeB temperature stability issue can be resolved by dynamically tracking field drift using navigator signals during scanning and subsequently correcting k-space data before image reconstruction. Note that NdFeB material is also significantly cheaper[60]. Our present 0.055 T magnet used about ~88 kg SmCo material. It cost ~USD10k, which can constitute >50% of total material costs, if our 0.055 T ULF brain MRI scanner is designed for low-cost quantity production. Second, we predict the ULF MRI scanner will be ideally light and mobile to facilitate point-of-care applications. Our present prototype scanner was relatively heavy with magnet assembly weighting ~750 kg though we expect that it can be readily reduced to ~500 kg or less through design optimization based on our preliminary simulations. We expect that future development can adopt the use of lighter homogenous cylindrical Halbach magnet designs[19,20], but preferably with relatively large inner magnet diameter and clear frontal openings (facing subject's face) for patient comfort. Last, we chose to develop a brain ULF MRI scanner in this study given the immense need and value of MRI in the diagnosis and prognosis of various neurological diseases and injuries. Presently, ~30% of clinical MRI cases involve the brain[61], and MRI remains the undisputed imaging modality for the brain due to its versatility, soft tissue sensitivity and specificity. Nevertheless, it is noteworthy that most ULF MRI technologies can be easily scaled to imaging other body parts, including extremity imaging and whole-body imaging.

In conclusion, we developed an ultra-low-field brain MRI scanner. It is completely shielding-free, low-cost, and potentially mobile. We succeeded in implementing four essential standard clinical imaging protocols on this hardware platform. We demonstrated the preliminary clinical feasibility in diagnosing tumor and stroke. The development of such ULF MRI technologies will enable patient-centric and site-agnostic MRI scanners to fulfill the unmet clinical needs across various global healthcare sites and has the potential to democratize MRI for low and middle income countries.

## Methods

**Magnet design**. Key magnet components were designed using electromagnetic field modeling to provide a homogenous 0.055 T field for adult brain imaging with adequate shoulder access (Fig. 1). Such homogenous field was achieved with additional passive shimming utilizing small iron and/or SmCo pieces through 3D field mapping. The resulting magnet dimensions were 95.2 cm, 70.6 cm and 49.7 cm (width x length x height) with a 30 cm clear vertical gap. The final 0.055 T field, corresponding to 2.32 MHz proton resonance frequency, had an inhomogeneity <250 ppm peak-to-peak over 240 mm DSV. Five Gauss fringe field was within 45 cm, 90 cm and 80 cm from the center of the magnet along width, length and height direction (Fig. 1c). Note that no external RF and magnetic shielding cages were needed. As such the scanner was compact and only occupied a footprint of ~2m², including both magnet and electronic cabinet.

Rare earth material Samarium-Cobalt with low temperature coefficient (Sm$_2$Co$_{17}$-22LTC) was chosen for construction of the main magnet (remanence Br = 0.95 T, BH$_{max}$ = 22 MGOe, coercivity H$_c$ = 680 kA/m, intrinsic coercivity H$_{ci}$ = 1600 kA/m) despite its lower BH$_{max}$ than NdFeB (BH$_{max}$ = 35-50 MGOe). SmCo offers dramatically improved temperature stability due to its extremely low temperature remanence coefficient (SmCo 0.015%/°C vs. NdFeB 0.125%/°C[27,28]). SmCo temperature coefficient of coercivity is also significantly lower (SmCo 0.15%/°C vs. NdFeB 0.45%/ °C[27,28]), thus offering more reproducible field with regards to temperature in presence of gradient pulsing during imaging. Moreover, SmCo has good mechanical strength and an excellent corrosion resistance without any special coating, ensuring good operational resilience against deformation, particularly during transport and movement as a mobile structure. Other key magnet components included an iron yoke (iron 99.9% pure), polar pieces (iron 99.9% pure), anti-eddy current plates made from silicon steel (DW310-35), and passive shimming rings made from low carbon steel (Q235).

**Gradient and radiofrequency (RF) subsystems**. The target field method based on the equivalent magnetic dipole approach was utilized to design the G$_x$, G$_y$, and G$_z$ gradient coils. The gradient coils were made using rectangular enameled wire and fixed onto epoxy resin boards to secure their respective winding patterns. G$_x$ and G$_y$ gradient coils were unshielded, whereas G$_z$ coil was actively shielded. The resistances of G$_x$, G$_y$, and G$_z$ coils were 47.0 mΩ, 45.5 mΩ and 75.5 mΩ, and their inductances were 120.1 μH, 98.9 μH and 88.2 μH, respectively. The sensitivities of G$_x$, G$_y$, and G$_z$ coils were 17.3 mT/m/100 A, 17.5 mT/m/100 A and 12.2 mT/m/100 A, respectively. The non-linearity of the gradient field was within 5% over 240 mm DSV with maximum gradient of 15mT/m. Eddy currents generated during gradient

pulsing were reduced using the anti-eddy plates. These plates were made from silicon steel and stacked on top of $G_z$. Gradient coils were driven by a PCI GA150 gradient amplifier (100 $V_{DC}$, 75 $A_{RMS}$ and 150 $A_{pk}$) (Performance Control Inc.).

Separate transmit and receive coils were employed. The RF transmit coil had a solenoid structure. At 0.055 T, transmit coil was typically driven by extremely low RF power. For example, the non-selective 1 ms 180° block pulse only required ~11 W peak RF power for brain imaging. RF receive coil was a one-channel room temperature solenoid coil with an ellipse cross-section (vertical axis 23.0 cm and horizontal axis 19.0 cm) and conventional designs[56,62] (10 winding turns and 9.5 cm length). Q factor of receive coil was measured to be approximately 30 and 31 when loaded and unloaded, respectively, offering adequate bandwidth for typical MRI signal (i.e., up to 50 kHz). A decoupling circuit was also implemented to detune the receive coil during RF transmission. RF signal was passed through a two-stage preamplifier module (first-stage: Gain = 30 dB; second stage: Gain =30 dB, for input $V_{PP} < 60$ mV). In addition, ten small resonant EMI sensing coils were fabricated with diameter of 5 cm and resonant frequency of 2.32 MHz. Three were placed in the vicinity of the patient head holder, four underneath the patient bed on patient left and right side, and three inside the electronic cabinet (near console, gradient amplifier and RF amplifier, respectively) as illustrated in Fig. 2a. These sensing coils were used to detect EMI signals only that were from both external environment and those generated internally by console/gradient/RF electronics during MRI scanning. Gradient and RF subsystems and data acquisition were controlled by a PC-based multi-channel NMR spectrometer console (EVO Spectrometer with Powerscan™ v6.3 software; www.mrsolutions.com).

**Deep learning driven EMI detection and elimination**. Within each repetition time (TR) during scanning, the main MRI receive coil and EMI sensing coils were used to simultaneously sample data within two windows, one was for the conventional MRI signal acquisition, the other was chosen for acquiring the EMI characterization data in absence of any MRI signals (i.e., EMI signals only) (Fig. 2b). For each scan, datasets sampled by both MRI receive coil and EMI sensing coils within the second window (i.e., EMI characterization acquisition) contained no MRI signal. After each scan, they were used to train a five-layer CNN model that could relate the 1D temporal EMI signal received by MRI receive coil to the 1D temporal signals received by multiple EMI sensing coils during EMI characterization window for each FE line (Fig. 2c). Note that the split for the datasets utilized here were 85% for training and 15% for validation. This model was then applied to the datasets sampled within MRI signal acquisition window at testing stage, where we could reliably predict the 1D EMI signal component in the 1D MRI receive coil signal based from the EMI signals simultaneously detected by the EMI sensing coils for each FE line. Subsequently, this EMI signal component was subtracted from the MRI receive signal, producing an EMI-free 1D MRI signal for that FE line. We applied this EMI cancellation procedure to all individual FE lines detected by MRI receive coil within MRI signal acquisition window before any signal averaging and subsequent image reconstruction (Fig. 2d). Note that each layer within the CNN model was a combination of convolution, batch normalization and rectified linear unit (ReLU), except the last layer where convolution operation only was performed. The kernel sizes of the five convolutional layers were 11 × 11, 9 × 9, 5 × 5, 1 × 1, and 7 × 7, respectively, with the corresponding number of channels being 128, 64, 32, 32, and 2. The complex data were processed by feeding the real and imaginary parts into the network as two separate channels. The input of the network was a 3D matrix with a size of $N_X \times 10 \times 2$, where $N_X$, 10 and 2 are number of points in one FE line, number of EMI sensing coils utilized, and number of channels corresponding to real and imaginary parts of the raw data. The output of the network was a 2D matrix with a size of $N_X \times 2$. The loss function was mean squared error (MSE). During training, MSE loss was minimized using Adam optimizer[63] with β1 = 0.9, β2 = 0.999 and initial learning rate = 0.0005. This deep learning procedure was implemented with a batch size of 16 for 20 epochs on a Quadro RTX 8000 graphics processing unit (GPU) and Intel Core i9-10900X central processing unit (CPU) using PyTorch 1.8.1 package. The typical training time for each scan protocol dataset was around 5 mins for T1W, T2W and FLAIR, and 20 mins for DWI. We found that this deep learning based EMI detection and elimination scheme was highly robust with regards to various external and internal experimental EMI sources for both phantom and human brain imaging (Fig. 3, and Supplementary Figs. 1 and 2). Note that, as shown in Fig. 2b, a time-overhead was introduced by the EMI signal characterization acquisition, which could increase the shortest possible TR in practice.

**ULF MRI scan protocols and optimization**. 3D GRE and FSE, and 2D EPI DWI sequences were first optimized on the low-cost 0.055 T hardware platform to minimize the effects of gradient eddy currents, main field inhomogeneity and drift, and low SNR. T1W and T2W scan protocols were acquired with 3D gradient-echo (GRE; TR = 52 ms, echo time TE = 13 ms, flip angle FA = 40°, acquisition matrix = 128 × 128 × 32, field-of-view FOV = 250 × 250 × 320 mm³, acquisition slice thickness = 10 mm, and number of excitations NEX = 2) and 3D fast spin echo (FSE; TR/TE = 1500/202 ms, FA = 90/180°, echo train length ETL = 21, acquisition matrix = 128x126x32, FOV = 250 × 250 × 320 mm³, acquisition slice thickness = 10 mm, and NEX = 2). Their scan times were approximately 5.5 mins and 7.5 mins, respectively. FLAIR scan protocol parameters were similar to those for T2W protocol above with FSE parameters TR/TE = 500/129 ms, ETL = 13,

acquisition matrix = 128 × 117 × 32, NEX = 4, and scan time 7.5 mins. Note that, instead of using the conventional inversion recovery approach, we suppressed the CSF here by simply using short TR saturation. With short TR of 500 ms at 0.055 T, CSF was dramatically suppressed due to its very long T1, while gray and white matter tissue signals were much less affected because of their very short T1s, producing a CSF-attenuated or FLAIR-like contrast. For both 3D GRE and FSE sequences above, elliptic 2D phase encoding patterns were used to reduce total scan time. DWI scan protocol was implemented with a 2D spin-echo EPI using a pair of diffusion gradients. The parameters were TR/TE = 2800/102 ms, acquisition matrix = 64 × 64, FOV = 250 × 250 mm², acquisition slice thickness/slice gap = 10/0 mm, diffusion time/duration Δ/δ = 49/30 ms, NEX = 52 for DWI images with b = 0 (i.e., b0 images) and images with b = 500 s/mm² (i.e., b1 images) diffusion weighting along three orthogonal directions. Total scan time was approximately 9.5 mins. For DWI, both EPI Nyquist ghosts and field inhomogeneity related geometric distortions were corrected when reconstructing b0 and b1 images. The final isotropic b1 images were combined from the 3 sets of b1 images with orthogonal diffusion weighting directions. ADC maps were calculated from b0 and b1 images. All image reconstruction procedures above were based on Fourier transform of fully sampled data. All images were reconstructed to 1 × 1 × 5 mm³ display resolution by applying zero padding in k-space.

**Study participants and clinical 3 T MRI scans**. The study was conducted under an institutional review board research protocol approved by The University of Hong Kong/Hospital Authority Hong Kong West Cluster (HKU/HA HKW IRB). A total of 34 out-patients were recruited for this study from neurology and neurosurgery clinics at Queen Mary Hospital, Hong Kong. Six patients dropped out of the study due to deteriorating medical condition before scheduled scans and a further three patients were excluded due to chest access issues at the 0.055 T scanner. Of the remaining twenty-five patients, thirteen were brain tumor, eight were subacute to chronic stroke and four were subacute to chronic intracerebral hemorrhage patients. Patients were screened for eligibility based on admission diagnosis, clinical examination, and the need for a clinical MRI follow-up examination. Exclusion criteria included at least one of the following contraindications to conventional MRI exam: cardiac pacemakers or defibrillators, intravenous medication pumps, insulin pumps, deep brain stimulators, vagus nerve stimulators, cochlear implants, pregnancy, claustrophobia and cardiorespiratory instability. We also recruited healthy volunteers for protocol optimization tasks at 0.055 T in this study. Written informed consent was obtained from all patients and healthy volunteers in the study prior to any MRI examination. All subjects that enrolled in the study also provided informed consent for publication of the images presented in all main and supplementary figures.

Patients were examined by 0.055 T brain ULF MRI scanner and a clinical GE 3 T MRI scanner (Signa Premier) using standard clinical neuroimaging protocols at Diagnostic Radiology at The University of Hong Kong. Whenever necessary, multi-slice 3 T images were rendered slightly to precisely match the 0.055 T multi-slice image orientations for direct comparison. Both 0.055 T and 3 T images were read on the same day by one senior clinical radiologist, (co-author H.K. Mak). The 0.055 T images were read and evaluated first while blinded to the corresponding 3 T images. The respective patient's clinical history was made available as part of standard clinical MRI reading procedure. The primary criteria for image evaluation were to determine whether and what specific lesions could be observed in the 0.055 T images. In this study, the 0.055 T images were not used for making any diagnostic or/and subsequent therapeutic decisions for the patients studied.

Using the standard clinical protocols, both the 3 T T1W and T2W images were acquired with 2D FSE (T1W: TR/TE = 2700/25 ms, inversion time = 830 ms, FA = 111°, ETL = 8, acquisition matrix = 340 × 280, FOV = 230 × 230 mm², acquisition slice thickness/slice gap = 5/0.5 mm, 27 slices, and NEX = 1; T2W: TR/TE = 5900/106 ms, FA = 120°, ETL = 30, acquisition matrix = 448 × 448, FOV = 230 × 230 mm², acquisition slice thickness/slice gap = 5/0.5 mm, 27 slices, and NEX = 2). The 3 T FLAIR images were acquired with 3D FSE with TR/TE = 6300/104 ms, inversion time = 1800 ms, FA = 90/180°, ETL = 180, acquisition matrix = 256 × 256 × 60, FOV = 250 × 250 × 150 mm³, acquisition slice thickness = 2.5 mm, and NEX = 1. The 3T DWI images were acquired with a 2D spin-echo EPI using a diffusion gradient pair. The parameters were TR/TE = 4000/57 ms, acquisition matrix = 120 × 160, FOV = 230 × 230 mm², acquisition slice thickness = 5 mm, 54 slices, NEX = 4 for images with b = 0 (i.e., b0 images) and images with b = 1000 s/mm² (i.e., b1 images) diffusion weighting along three orthogonal directions. The total 3 T scan time was ~20 mins for T1W, T2W, FLAIR and DWI, with acquisition resolution 0.7 × 0.8 × 5.0 mm³, 0.5 × 0.5 × 5.0 mm³, 1.0 × 1.0 × 4.0 mm³, and 1.9 × 1.4 × 5.0 mm³, respectively.

Note that the 0.055 T scanner was developed to explore and demonstrate its potential for human brain imaging only. Use of the methods and designs reported for phantom, human and animal imaging should be subject to the approval by relevant local safety, ethical and medical authorities.

**Acoustic noise level generated at 0.055 T**. Recordings of the SPL of acoustic noise levels generated by the brain ULF MRI scanner were made using an omnidirectional condenser microphone (M50, Earthworks) and a digital field recorder (DR-680, Tascam). Microphone was placed in the isocenter of MRI receive coil inside magnet. Recordings were made across six conditions with no subject inside

magnet (i.e., when the scanner was off, on, and scanning with T1W, T2W, FLAIR and DWI protocols). To facilitate the comparison with acoustic noise level measurements at 3 T made in previous studies[35,39,64], recordings of SPL made (in dB) at 0.055 T were further processed by applying an A-weighted filter (reported as dBA). In the A-weighted filter[35,39,64], the decibel values of sound intensity at low frequencies (<1 kHz) and high frequencies (>6 kHz) are reduced to reflect the varying sensitivity of the human ear to sounds at different frequencies within the normal hearing range (20 Hz-20 kHz).

**Reporting summary**. Further information on research design is available in the Nature Research Reporting Summary linked to this article.

## Data availability

All main data used, analyzed and generated that support the findings of this study, and key technical documents are available for download from a public repository (https://github.com/bispmri/Ultra-low-field-MRI-Scanner). The data for the spectral plots in Fig. 3 and Supplementary Fig. 4 are provided with this paper as a source data file. Other information is available from the corresponding author upon reasonable request. Source data are provided with this paper.

## Code availability

The custom software codes used for the EMI removal demonstrated in Fig. 3 can be also downloaded from the public repository (https://github.com/bispmri/Ultra-low-field-MRI-Scanner). Other information is available from the corresponding author upon reasonable request.

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

## Acknowledgements
This work was supported by Hong Kong Research Grant Council (R7003-19F, HKU17112120 and HKU17127121 to E.X.W., and HKU17103819, HKU17104020 and HKU17127021 to A.T.L.L.), Lam Woo Foundation, and Guangdong Key Technologies for Treatment of Brain Disorders (2018B030332001) to E.X.W. We would like to thank Dr. X. Ma, Messrs. C. Man, V. Lau, C. Ho, Z. Yi, J. Hu, S. Su, Z. Huang, E. Hui, and Ms. N. Hou and L. Xie for their technical assistance, Drs. S. Lau, B. Taw, K. Cheng, L. Li, J. Zhuang and the clinical teams at the Neurosurgery Division of Department of Surgery and Neurology Division of Department of Medicine, Queen Mary Hospital for their clinical advice, and Ms. J. Chau, R. Liu and V. Sin for their assistance in handling the logistics of patient recruitment. We would also like to thank Drs. P. Khong, W. Chew, J. Gore, P. van Zijl, G. Pang, B. Rosen, and K. Chan for discussions.

## Author contributions
Y.L., A.T.T.L. and E.X.W. contributed to or advised on system and method design and development. Y.L., A.T.T.L., Y.Z., L.X. and E.X.W. contributed to the technical development and optimization of 0.055 T protocols. H.K.F.M., A.C.O.T., G.K.K. Lau, and G.K.K. Leung. advised on the design of experiments to demonstrate initial clinical feasibility, recruited patients, and provided guidance for interpretation of images acquired at ultra-low-field and 3 T clinical MRI. A.T.L.L. and E.X.W. wrote the manuscript. All authors contributed to reviewing and editing the manuscript.

## Competing interests
The authors declare no competing interests.
