## [Peer Review File · Nature Communications]

A Low-cost and Shielding-free Ultra-low-field Brain MRI ScannerEditorial Note: Parts of this Peer Review File have been redacted as indicated to maintain the confidentiality of unpublished data.

REVIEWER COMMENTS

Reviewer #1 (Remarks to the Author):

Review for NCOMMS-21-28454: "A Low-cost and Shielding-free Brain MRI Scanner for Accessible Healthcare"

I agree to these reviewer comments being published along with the paper.

Signed: Florian Knoll, September 3rd 2021, Erlangen, Germany

Summary of the manuscript:

This manuscript describes the design and development of a low-field (0.055T) permanent magnet MRI scanner for brain imaging. The scanner is described as a low-cost, low-maintenance system that runs via a regular AC power socket, due to its permanent magnet design does not require liquid helium, weighs 750kg and does not require a dedicated shielded room for operation.

A particular focus of the described methodological novelty is on a procedure to remove external electromagnetic interference (EMI) during a scan, which is a prerequisite to operate the scanner without an RF-shield. The approach is based on the addition of 10 EMI sensing coils that are added to the system in addition to the usual RF transmit and receive coils. These coils have the purpose to pick up the electromagnetic interference during a scan, and a deep learning model is then used to predict the interference signal that is generated in the MRI receive coil based on these measurements. This predicted interference signal is then subtracted from the actual MRI signal, which performs a correction of the external interference. It is demonstrated in phantom and in-vivo-scans of healthy volunteers that this procedure leads to a substantial improvement of image quality when scans are performed without an RF-shield.

The manuscript then demonstrates the acquisition of four image contrasts that are important for clinical brain scans (T1w, T2w, FLAIR and DWI). An imaging protocol based on these contrasts is tested in a clinical study with 25 patients with a range of pathology, which were scanned both on a conventional 3T clinical MRI scanner, and the developed 0.055T scanner.

Finally, the manuscript provides an outlook towards potential benefits of the 0.055T scanner, in particular a phantom scan of metal implants that demonstrates the reduced susceptibility artifacts at this field strength, the reduced acoustic noise during data acquisition, and the ability to acquire an additional image contrast (TrueFISP).

Critique:

This is an excellent manuscript, and the result of a tremendous research effort by the authors. Low field MRI is an upcoming research area in the community because the limited availability of MRI outside of specialized radiology departments in high-income countries is a big challenges in the field. While this is not the first low-field permanent magnet design that is proposed in the literature, the scanner design is convinving. The proposed EMI cancellation approach is novel, the corresponding results look convincing and the possibility to operate the scanner outside a dedicated RF-shield is an important step towards scanner deployment in low-resource settings.

The clinical demonstration of the most important brain image contrasts is a strong feature of a manuscript where the core effort was a new methodological hardware development. The resulting images are of course not of a quality that we tend to expect to see from clinical systems, but as the authors point out, the goal of this 0.055T system is not to replace current state-of-the-art high-field scanners. It is rather to provide a new diagnostic tool that is used at a different stage in the diagnostic chain, or in situations where no images would otherwise be available at all. Therefore the image quality has to be judged in such context.

My main critique is that given there is a strong focus towards the methodological development for EMI removal, I found that the manuscript was lacking background and literature discussion of this

area, to motivate the importance of this development. The ability to perform scans without an RF-shield is a strong asset, but to my knowledge other low-field scanners (e.g. the commercial scanner from the company hyperfine that the authors mention briefly in the introduction) also operate without an RF-shield. It would be interesting to provide context how the proposed approach is either different or similar to existing approaches for EMI artifact removal that have been presented in the field.

In addition to this main comment, I have several other comments that I believe will further improve the manuscript.

Specific Comments:

1) The description of the proposed EMI cancellation approach would benefit from some additional details. In particular:

1a) It is claimed that "the deep learning EMI cancellation procedure is able to eliminate the undesirable EMI signals in a highly reliable and robust manner even when environmental EMI sources and their spectral characteristics changed dynamically during scanning". This is a strong claim and indeed essential for real-world deployment of the scanner. However, I did not see any experiments and results that specifically tested the impact of EMI that changed dynamically during a scan. This could be done either with a simulation, where EMI is introduced only during short randomly selected periods during the scan, and/or with an experiment with an EMI source that can be turned on and off during the scan.

1b) Since periods need to be interleaved in the pulse sequence where the EMI signal is detected, which additional constraints does this impose on the design of the pulse sequence? Can you discuss the time-overhead that is introduced by this? I'm not sure if the time-axis in Figure 2 is scaled consistently, but it looks like half of the time during each TR is essentially wasted for data acquisition because it is used for EMI detection. In particular in the context of low field imaging, with the substantially shorter T1 times mentioned by the authors, I expect that sequences will benefit from much shorter TRs in comparison to current clinical protocols, and the time-window to detect the EMI will essentially double the minimal TR that can be achieved.

1c) Please provide more details about the training of the neural network, and the exact architecture of the convolutional neural network. Regarding the training, was each frequency encoding line considered to be a data sample, and do I understand it correctly that an individual neural network was trained for each MRI scan? In that case, how was the split into training, validation and test data performed? How were hyper parameters selected? Which optimization algorithm was used for training and which convergence criterium was used?

1d) It is claimed that the EMI cancellation procedure "obtained phantom and brain image SNRs that were $\geq 95\%$ of those when using a fully enclosed RF shielding cage for direct comparison." Since no comparison to an acquisition with an RF-shield was performed, this statement should be removed.

2) The brain image results appear to have almost no contrast between gray and white matter. I understand that the results cannot be compared to what we are used to seeing from current clinical systems, but given that the superior soft-tissue contrast is one of the main assets of MRI over modalities like CT, I feel this should be addressed. Is this an inherent limitation at this field strength because the T1 and T2 values are much closer together, or is it just a matter of lower contrast to noise ratio? If the reason is contrast to noise ratio, this could be tested in an experiment with a higher number of averages to boost SNR.

3) It is stated that "Scanning was acoustically much quieter when compared to high-field MRI, with maximum peak sound pressure level (SPL) < 85 dBA at 0.055 T (Supplementary Fig. 1) vs. ≤ 120 dBA at 3T." I do not doubt that the scanner creates less acoustic noise, but the statement suggests that the an actual measurement of the noise level was performed at 3T as well, with

matched experimental settings and pulse sequences. Unless the noise measurement was done exactly in the same way and with the same sequences as in (30), I suggest tone down this statement.

4) Sequence parameters:

4a) Please provide the the echo train length of the 3D fast spin echo acquisitions.

4b) Please clarify how the reconstructed image resolution of $1 \times 1 \times 5$ mm³ was achieved and why this step was performed. Was this done with k-space zero padding, and was the reason to make the images more comparable to 3T for the clinical study?

4c) Please report sequence parameters for the standard clinical 3T protocols as well.

5) Please provide more details about the clinical study. Was the radiologist fully blinded to the corresponding 3T images and the clinical presentation of the patients, or did he/she read those as well at a different point in time, or was this done by a different radiologist? Was the only criterium presence/absence of pathology? Were there cases where the diagnosis on 3T and 0.0055T did not match, and in that case would that have lead to a different therapeutic decision?

6) Please indicate the direction of the B₀ field in Figure 8.

7) p11: To some degree, the section "Promises of imaging at ultra-low-field (p11)" is redundant to the introduction. This should be streamlined.

8) It is mentioned that at ULF, the noise in MRI signals is dominated by the RF receiver coil noise, while the sample noise is negligible. Did that influence the design of the RF coil that was built for the scanner?

9) I assume that a total of 34 patients were recruited for the study, since 6+3 ended up not being included, and the total for the study was still 25 (page 20 lines 472 to 475).

10) I appreciate that data will be made available publicly and custom computer codes will be made available from the corresponding author upon request. In the spirit of reproducible research, I would appreciate it even more if those were included in the public repository as well. The strongest impact would of course be achieved if the authors also shared hardware plans and component lists, which would allow other research groups to reproduce the authors' design.

Reviewer #2 (Remarks to the Author):

This paper is well-written and presents the highest quality low-field MR images that I have seen. Certainly they are a massive improvement on the highly distorted images shown in a previous publication in Nature Communications. An impressive degree of EMI reduction is shown with the neural network, and images using highly B₀-sensitive sequences such as EPI and TrueFISP are presented for the first time.

In terms of science the main novelty is the means of EMI reduction. Certainly the ability to operate such a system in an RF noisy environment is critical, but several other groups have shown that very simple shielding can probably suffice, so it is unclear whether this is a true breakthrough. For example, O'Reilly et al. have shown that a simple conductive cloth eliminates the vast majority of EMI, and the Hyperfine unit also incorporates a (commercially proprietary) highly effective EMI reduction scheme.

There are a few references that should be added, for example the issue of reduced artifacts from metallic implants has been covered by van Speybroeck et al in 2021.

RESPONSES AND REVISION OF NC MANUSCRIPT # NCOMMS-21-28454

A Low-cost and Shielding-free Brain MRI Scanner for Accessible Healthcare

Reviewer #1:

R1-G (Summary) This manuscript describes the design and development of a low-field (0.055 T) permanent magnet MRI scanner for brain imaging. The scanner is described as a low-cost, low-maintenance system that runs via a regular AC power socket, due to its permanent magnet design does not require liquid helium, weighs 750 kg and does not require a dedicated shielded room for operation.

A particular focus of the described methodological novelty is on a procedure to remove external electromagnetic interference (EMI) during a scan, which is a prerequisite to operate the scanner without an RF-shield. The approach is based on the addition of 10 EMI sensing coils that are added to the system in addition to the usual RF transmit and receive coils. These coils have the purpose to pick up the electromagnetic interference during a scan, and a deep learning model is then used to predict the interference signal that is generated in the MRI receive coil based on these measurements. This predicted interference signal is then subtracted from the actual MRI signal, which performs a correction of the external interference. It is demonstrated in phantom and in-vivo-scans of healthy volunteers that this procedure leads to a substantial improvement of image quality when scans are performed without an RF-shield.

The manuscript then demonstrates the acquisition of four image contrasts that are important for clinical brain scans (T1w, T2w, FLAIR and DWI). An imaging protocol based on these contrasts is tested in a clinical study with 25 patients with a range of pathology, which were scanned both on a conventional 3T clinical MRI scanner, and the developed 0.055 T scanner.

Finally, the manuscript provides an outlook towards potential benefits of the 0.055 T scanner, in particular a phantom scan of metal implants that demonstrates the reduced susceptibility artifacts at this field strength, the reduced acoustic noise during data acquisition, and the ability to acquire an additional image contrast (TrueFISP).

We sincerely thank the reviewer for his concise *and* precise summary of the core aspects of our study.

R1-G (General Comments)

R1-G-1. This is an excellent manuscript, and the result of a tremendous research effort by the authors. Low field MRI is an upcoming research area in the community because the limited availability of MRI outside of specialized radiology departments in high-income countries is a big challenge in the field. While this is not

the first low-field permanent magnet design that is proposed in the literature, the scanner design is convincing. The proposed EMI cancellation approach is novel, the corresponding results look convincing and the possibility to operate the scanner outside a dedicated RF shield is an important step towards scanner deployment in low-resource settings.

We thank the reviewer for sharing his enthusiasm for our work.

R1-G-2. The clinical demonstration of the most important brain image contrasts is a strong feature of a manuscript where the core effort was a new methodological hardware development. The resulting images are of course not of a quality that we tend to expect to see from clinical systems, but as the authors point out, the goal of this 0.055 T system is not to replace current state-of-the-art high-field scanners. It is rather to provide a new diagnostic tool that is used at a different stage in the diagnostic chain, or in situations where no images would otherwise be available at all. Therefore, the image quality has to be judged in such context.

We thank the reviewer for sharing his enthusiasm for our work.

Ultra-low-field (ULF) MRI is a rapidly developing field at present time, particularly given the healthcare disparities that exist in many parts of the world. We shall share the key technical design details, codes and data so to inspire both traditional MRI and other research communities (electromagnetics, materials, computing and data science). We hope our research results and information sharing here will galvanize low-cost MRI technology development for point-of-care clinical applications, and for MRI accessibility in the developing and underdeveloped world.

R1-G-3. My main critique is that given there is a strong focus towards the methodological development for EMI removal, I found that the manuscript was lacking background and literature discussion of this area, to motivate the importance of this development. The ability to perform scans without an RF shield is a strong asset, but to my knowledge other low-field scanners (e.g., the commercial scanner from the company Hyperfine that the authors mention briefly in the introduction) also operate without an RF shield. It would be interesting to provide context how the proposed approach is either different or similar to existing approaches for EMI artifact removal that have been presented in the field.

- i. We thank the reviewer for drawing our attention to this issue concerning the background literature and why we decided to develop a deep learning driven electromagnetic interference (EMI) detection and cancellation scheme, and the differences/similarities to other existing/ongoing approaches taken by others so far. We agree that more technical background, method details and discussions regarding our EMI elimination strategy should be provided in this manuscript.

- ii.* We started tackling this active EMI elimination challenge for low-cost and radiofrequency (RF) shielding-free MRI five years ago by first defining the key requirements as follows. Our key considerations for EMI removal strategy are as follows: **(a)** The developed method must achieve effective and nearly total EMI removal when compared to the ground truth scenario (, i.e., when an RF shielding cage is deployed to fully enclose the subject during MRI scan). **(b)** The method must be able to deal with EMI signals that changes dynamically over time during MRI scan. In reality, such changes can arise from the surrounding EMI sources with various nature and behaviors. EMI signal received by MRI receive coil can be also influenced by the human body, which serves as an effective antenna^{1,2} for EMI reception in a shielding-free MRI setting. For example, varying body size and weight can alter the level and characteristics of EMI signal picked up by body and subsequently detected by MRI receive coil. Human body position change during MRI scan can also alter the EMI signal detected by MRI receive coil due to the change in electromagnetic coupling between surrounding EMI emitting sources and receiving human body. **(c)** The EMI signal from scanner internal low-cost MRI electronics (e.g., gradient and RF amplifiers, console, power supplies, and their suboptimal insulations) must be dealt with as well. *Therefore, an active, accurate, highly resilient and relatively simple EMI removal strategy is highly desired for RF shielding-free and low-cost ULF MRI. Such successful strategy is a mandate if one wishes to implement the intrinsic low-SNR MRI protocols at ULF such as diffusion-weighted imaging, a key clinical neuroimaging protocol.*
- iii.* We subsequently solved this active EMI detection and cancellation problem for RF shielding-free MRI by **(a)** taking advantage of the well-established multi-receiver MRI electronics (previously developed for parallel imaging) and **(b)** utilizing separate resonant or tuned EMI sensing RF coils to simultaneously acquire EMI signals in various spatial locations during two windows, namely, MRI signal acquisition and EMI signal characterization windows (as illustrated in main Fig. 2). We derive and establish a model to predict the EMI signal in MRI receive coil from the EMI signals detected by EMI sensing coils. Intuitively, an accurate prediction model should be highly feasible because of a simple electromagnetic phenomenon. That is, the properties of RF signal propagations among any radiative (e.g., air) or/and conductive media (e.g., surrounding EMI emitting structures such as power lines and nearby equipment, RF coils, other MRI hardware pieces and cables, patient bed, imaging object or human body) are fully dictated by the electromagnetic coupling among these media or structures. Such coupling relationships can be analytically characterized in a relatively simple manner by the frequency domain coupling or transfer functions among structures (e.g., among MRI receive coil and sensing coils).
- iv.* In our work, given the practical considerations outlined in **R1-G-3-ii** above, we opted to develop a deep learning driven approach to derive this prediction model (instead of other adaptive³ or analytical approach⁴) for establishing the relationships among the EMI signals detected by EMI sensing coils and MRI receive coil. *We hypothesized that the nature of deep learning should enable a more accurate*

and robust EMI prediction and cancellation procedure, yet relatively simple for applications in diverse and unshielded imaging environments. Indeed, this belief is now well supported by our results in both original submission and revised submission here.

- v. As pointed by the reviewer, commercial company Hyperfine has recently demonstrated a 0.064 T head MRI scanner without RF shielding room requirement (www.hyperfine.io/portable-mri) using an undisclosed and proprietary EMI removal method. Although its details (as well as other key information regarding scanner hardware and imaging sequences) are unavailable, we speculate that it is based on the use of frequency domain transfer function approach and EMI sensing coils⁵.

We're also aware that three other academic groups have been actively seeking solutions to mitigate EMI issues for imaging in unshielded environments. In 2018, one group proposed to use 3 orthogonal magnetometers and one 2nd order gradiometer to sense environmental EMI and then estimate the EMI signal in MRI receive coil for ULF through an adaptive estimation and suppression procedure for ULF imaging³. However, the results were suboptimal and the approach was hardware demanding.

Group from Leiden have attempted to use simple conductive cloth to cover the subject during scan⁶. This passive method can alter and reduce EMI signal mainly from external environments. *However*, its performance is far from optimal. It is also inadequate to deal with potential EMI sources from scanner internal electronics.

Group from Vanderbilt/MGH are presently working on analytical approaches to estimate and remove external EMI signal received by MRI receive coil using tuned EMI sensing RF coils (as reported in their preliminary studies during August 2020 ISMRM Annual Meeting⁴ and May 2021 ISMRM Annual Meeting⁷). Their first approach was based on the frequency domain transfer function concept as discussed in **R1-G-3-iii** and implemented in frequency domain but with only limited preliminary results for demonstration⁴. The second approach was also based on the frequency domain transfer function concept but implemented in the time domain as linear convolutions under the *assumptions* that their corresponding transfer functions (between MRI receive coil and EMI sensing coils in frequency domain) were relatively smooth within MRI signal receiving bandwidth *and* stable during scan (so finite and small convolution windows could be used for linear convolution operations, partly resembling the GRAPPA parallel imaging concept⁸), and EMI spectra should be generally bandlimited. This approach analytically modeled the relationship between EMI signals simultaneously detected by MRI receive coil and EMI sensing coils through time domain linear convolutions based on these specific assumptions, thus can be (a) potentially vulnerable to changes in the transfer functions during scan (e.g., due to patient body position change or movement of nearby attending staff or equipment as discussed in **R1-G-3-iii** above), (b) potentially problematic given that EMI spectra can be often broadband instead of bandlimited or compact within the MRI sampling bandwidth. The Vanderbilt/MGH group partly mitigated these issues (mostly issue a) by analyzing and subdividing

the dataset to sub-datasets to accommodate these EMI signal characteristic changes during scan. *However*, such adaptive procedure, given the assumptions regarding transfer functions (vs. the reality as discussed above), likely would limit the robustness of their proposed EMI cancellation strategy. Their preliminary results demonstrated substantial EMI removal *but required the use of electrodes mounted on body surface as well as conductive cloth to cover subject⁶* in order to achieve highly effective EMI removal, which we consider also cumbersome in practice. *Further*, the efficacy of this latter strategy also remains to be fully assessed through a cohort of subjects or/and patients (over 100 subjects/patients in our study so far instead of few subjects only in their study) in terms of removing both external and internal EMI without introducing artifacts. Note that we aimed to target and eliminate both external EMI and EMI from scanner internal electronics in our present study.

- v. In this work, we report a deep learning driven EMI prediction and cancellation strategy. We included more experiments and analyses. We demonstrated that our method worked robustly with dynamically varying EMI from both external environments and scanner internal electronics, and even in presence of subject body position change during scan (See main Fig. 3, **Figs. R1-1a-1 to R1-1a-5** below). More importantly, in this revision, we compared our EMI prediction and cancellation performance directly to the ground truth scenario, i.e., when a fully enclosed RF shielding cage was installed to cover the subject (See **Figs. R1-1d-1 and R1-1d-2** below). These results demonstrate the highly effective EMI removal performance by our proposed method, producing final image noise levels as low as those obtained using a fully enclosed RF shielding cage (*within 5% range*) in human brain experiments. Note that no conductive cloth and body EMI pickup electrodes (as mandated in the Vanderbilt/MGH approach⁷) were employed in our study.
- i. In this revision, as suggested by the reviewer, we have included statements and discussions regarding the present landscape of EMI removal strategies and why we chose to pursue a deep learning based approach as discussed above in **R1-G-3-v**. Please also see our detailed responses and results in **R1-1a** and **R1-1d** below.

These additions have been made to the introduction section and ‘*0.055 T brain ULF MRI system hardware design*’ subsection of the results section.

In addition to this main comment, I have several other comments that I believe will further improve the manuscript.

R1-1 The description of the proposed EMI cancellation approach would benefit from some additional details. In particular:

R1-1a It is claimed that "the deep learning EMI cancellation procedure is able to eliminate the undesirable EMI signals in a highly reliable and robust manner even when environmental EMI sources and their spectral characteristics changed dynamically during scanning". This is a strong claim and indeed essential for real-world deployment of the scanner. However, I did not see any experiments and results that specifically tested the impact of EMI that changed dynamically during a scan. This could be done either with a simulation, where EMI is introduced only during short randomly selected periods during the scan, and/or with an experiment with an EMI source that can be turned on and off during the scan.

- i. We thank the reviewer for drawing our attention to this claim made in the original submission without supporting data.
- ii. As discussed in **R1-G-3-ii**, **iii** and **iv** above, we chose to develop a deep learning driven method to predict and cancel the EMI signal detected by MRI receive coil because of the diverse and complex nature of various EMI sources (from both external environments and internal scanner low-cost electronics) in realistic shielding-free imaging settings. Our extensive experimental experience with shielding-free ULF MRI from the past five years has been that certain EMI sources indeed can vary slowly or rapidly over time during scan. Further, EMI signal picked up by human body^{1,2} can vary when subject changes body position during scan. Body position can influence EMI pickup levels and characteristics due to change of the electromagnetic coupling or transfer functions between EMI emitting sources and the receiving human body. Therefore, a data driven method such as deep learning is intuitively preferable over the analytical approaches for robustness and resilience under various shielding-free imaging settings.
- iii. The experimental data shown in main Fig. 3 did contain EMI signals from different sources that varied dynamically during scan. We apologize for not explicitly showing the temporal characteristics of the raw k-space data in our original submission. **Figs. R1-1a-1, 2, 3** and **4** below present the analyses of the raw data acquired under the three conditions that correspond to the main Fig. 3.

As shown in **Figs. R1-1a-1** and **R1-1a-3** below, in absence or presence of MRI signals (i.e., RF transmit power off or on) and before EMI elimination, one can easily discern and identify numerous sources of external (environmental or/and artificially generated) EMI and internal (e.g., from console) EMI. They were indeed not static across the duration of each scan. For example, it can be clearly seen from the frequency spectra (i.e., Fourier transform of frequency encoding line, or FT of FE) that spectral characteristics of some EMI sources changed dynamically, either slowly or very rapidly, over time during scan. *The results after EMI cancellation (shown in **Figs. R1-1a-2** and **R1-1a-4** below), together with the image and spectral results in Fig. 3, clearly demonstrate the effectiveness of the proposed deep learning EMI cancellation method in presence of diverse and dynamically varying EMI signals.*

Figure R1-1a-1. Spectral analyses of the raw 3D FSE T2-weighted data (corresponding to the spectral results shown in main Figure 3A-C, respectively, without EMI elimination, and with RF transmit power off) show the dynamically changing external and internal EMI sources. Note that, for illustration, only consecutive 1000 phase encoding (PE) samples, acquired over a ~72 s time span, were plotted along horizontal direction. Both FE line data and their Fourier transform magnitude (FT of FE) are shown. (A) Phantom data corresponding to main Figure 3A, with additional strong broadband EMI generated from a nearby source. (B) Phantom data corresponding to Figure 3B, with additional swept frequency EMI generated from a nearby source (center frequency = 2.32 MHz, sweep span = 100 kHz, frequency points = 101, and continuous sweeping cycle period = 4 s). Note that narrowband EMI from an internal scanner source (console) was also visible. (C) Human brain data corresponding to Figure 3C, with signals from various external EMI sources. Spectral characteristics of these EMI sources during scan changed rapidly or gradually (indicated by yellow arrows) in terms of amplitude and frequency. Note that temporal changes of EMI characteristics were not obvious in the phantom data shown in (A) because the EMI artificially generated by the nearby broadband source overwhelmed the background EMI. Also note that, more external EMI sources can be identified in (C) partly because human body acted as an antenna and picked up more EMI signal. Nevertheless, the proposed EMI cancellation strategy worked robustly in presence of these temporally changing EMI signals from both narrowband and broadband sources, as supported by the spectral results in Figure 3.

Figure R1-1a-2. Spectral analyses of the raw 3D FSE T2-weighted data (corresponding to the spectral results shown in main Figure 3A-C, respectively, with EMI elimination, and with RF transmit power off). All other descriptions are the same as those in Figure R1-1a-1. By comparing with Figure R1-1a-1, the proposed EMI cancellation strategy eliminated all EMI signals from various sources, as directly demonstrated by absence of any discernible EMI signals here. The results were displayed with enhanced brightness ($\times 3$) compared to Figure R1-1a-1 for easy visualization here.

Figure R1-1a-3. Spectral analyses of the raw 3D FSE T2-weighted data (corresponding to the image and spectral results shown in main Figure 3A-C, respectively, without EMI elimination, and with RF transmit power on), show both MRI signals and dynamically changing external and internal EMI sources. (A) Phantom data corresponding to Figure 3A, with additional broadband EMI generated from a nearby source. (B) Phantom data corresponding to Figure 3B, with additional swept frequency EMI generated from a nearby source. Note that narrowband EMI from an internal scanner source (console) was also visible. (C) Human brain data corresponding to Figure 3C, with signals from various external EMI sources. All other descriptions are the same as those in Figure R1-1a-1. Similar to Figure R1-1a-1, various EMI sources could be observed, some of which varied dynamically over time during scan. Again, more external EMI sources could be identified in (C) partly because human body acted as an antenna and picked up more EMI signal. The proposed EMI cancellation method worked robustly in presence of these dynamically changing EMI signals, both narrowband and broadband, as supported by both image and spectral results in Figure 3.

Figure R1-1a-4. Spectral analyses of the raw 3D FSE T2-weighted data (corresponding to the image and spectral results shown in main Figure 3A-C, respectively, with EMI elimination, and with RF transmit power on). All other descriptions are the same as those in Figure R1-1a-1. The proposed EMI cancellation strategy eliminated all EMI signals from various sources as shown in Figure R1-1a-3, as directly demonstrated by presence of MRI signals only and absence of any discernable EMI signals here (and supported by the spectral and images results in Figure 3). The results were displayed with the same brightness as that in Figure R1-1a-3.

- iv. To illustrate the effect of subject body position change on EMI signal detected by MRI receive coil during scan (as alluded in R1-1a-ii above), **Fig. R1-1a-5** below shows the four individual images acquired during a FLAIR protocol, before EMI removal, using the standard 4-average brain 3D FSE FLAIR acquisition described in the manuscript. It can be seen here that a moderate body position change in the middle of a scan (i.e., bending of lower legs between 2nd and 3rd acquisitions) caused changes in detected EMI characteristics (both frequency and amplitude), *demonstrating another scenario that can be encountered in realistic imaging settings due to subject position changes,*

movements of nearby attending staff or equipment during scan. Nevertheless, our proposed deep learning driven EMI cancellation strategy still effectively eliminated all EMI noise from the images.

Figure R1-1a-5. Changes in detected EMI signals due to a change of subject body position during the standard 4-average FLAIR scan. Four individual 3D FSE FLAIR image datasets were sequentially acquired before averaging from a normal adult. The corresponding images without EMI removal are shown on the left. Only four brain slices are shown in the four rows here. Frequency encoding (FE) was along vertical direction. Before EMI cancellation, external narrowband EMI could be seen as the horizontal noise bands located at the bottom of these individual images (as well as the thinner horizontal noise bands around the middle of the images). Yellow and green boxes indicate the EMI characteristics before and after the body position change, respectively. It can be seen that a moderate subject body position change, i.e., bending of legs between the 2nd and 3rd acquisitions, led to notable difference in the external narrowband EMI frequency location and magnitude (because of the change in electromagnetic coupling between surrounding EMI emitting sources and the receiving human body). This demonstrated another dynamically varying EMI scenario that can be encountered in realistic imaging setting due to subject position changes, or movements of nearby attending staff and equipment during scan. With the proposed deep learning driven EMI elimination strategy, EMI noise/artifacts, both narrowband and broadband EMI, were effectively removed as demonstrated by the final EMI-free images on the right.

- v. In this revision, we have included **Figs. R1-1a-1** and **R1-1a-2** above as the new **Supplementary Figs. 1** and **2** to illustrate the dynamically varying nature of the EMI signals in main Fig. 3, and revised the text accordingly.

R1-1b Since periods need to be interleaved in the pulse sequence where the EMI signal is detected, which additional constraints does this impose on the design of the pulse sequence? Can you discuss the time-overhead that is introduced by this? I'm not sure if the time-axis in Figure 2 is scaled consistently, but it looks like half of the time during each TR is essentially wasted for data acquisition because it is used for EMI detection. In particular, in the context of low field imaging, with the substantially shorter T1 times mentioned by the authors, I expect that sequences will benefit from much shorter TRs in comparison to current clinical protocols, and the time-window to detect the EMI will essentially double the minimal TR that can be achieved.

- i. We thank the reviewer for insightfully drawing our attention to this time-overhead issue introduced by the implementation of EMI signal characterization window in our proposed EMI cancellation method.
- ii. The time-axis in Fig. 2B is not scaled consistently to the 3D FSE pulse sequence. *In some scenarios*, the inclusion of the EMI signal characterization window can increase the repetition time (TR), in which case it will not exactly increase TR by two-fold because gradients and RF waveforms do not need to be repeated.
- iii. We agree that it is desirable to preserve the shortest possible TR for imaging flexibility at ULF. **(1)** For our 0.055 T 3D GRE T1W scan protocol, such flexibility is reduced to certain extent because ideally the shortest possible TR may need to be preserved for flexible acquisition efficiency or contrast optimization in the future. **(2)** For our 0.055 T 3D FSE T2W and 3D FSE FLAIR protocols, this time-overhead do not impose any limitation because their TRs are already relatively long. **(3)** For our 0.055 T 2D EPI DWI protocol where 2D multi-slice whole brain coverage is necessary, however, this time-overhead can partly compromise the acquisition efficiency. Short TR is desired in 2D whole-brain DWI at ULF for acquisition efficiency because TR only needs to be ~3 times the gray and white matter T1 values (which are short and all less than 400 ms at 0.055 T). The TR in our present 2D EPI DWI protocol was long (i.e., 2800 ms) to provide 12-slice coverage, partly due to this time-overhead issue.

[Redacted]

[Redacted]

[Redacted]

- v. In this revision, we acknowledge in the ‘*Deep learning driven EMI detection and elimination*’ subsection of the methods section that the TR of our imaging protocols are increased as a consequence of the proposed EMI cancellation strategy.

R1-1c Please provide more details about the training of the neural network, and the exact architecture of the convolutional neural network. Regarding the training, was each frequency encoding line considered to be a data sample, and do I understand it correctly that an individual neural network was trained for each MRI scan? In that case, how was the split into training, validation and test data performed? How were hyper parameters selected? Which optimization algorithm was used for training and which convergence criterium was used?

- i. The input to the CNN network was a 3D matrix with a size of $N_X \times 10 \times 2$. N_X , 10 and 2 represent the number of points in one frequency encoding (FE) line, the number of EMI sensing coils utilized in our system, and 2 channels (for real and imaginary parts of the raw data), respectively.

The CNN consisted of five layers. The respective kernel sizes of the five convolutional layers were 11×11 , 9×9 , 5×5 , 1×1 , and 7×7 with the corresponding number of channels being 128, 64, 32, 32, and 2. The output of the network was a 2D matrix with a size of $N_X \times 2$.

Yes, each frequency encoding line was considered as a data sample and network was trained for each MRI scan. We made this choice mostly for simplicity.

- ii. As described in the methods section of the original manuscript, within each TR during scanning, the MRI receive coil and EMI sensing coils were used to simultaneously sample data within two acquisition windows, one was for the conventional MRI signal acquisition, the other was for acquiring the EMI characterization data in absence of any MRI signals (main Fig. 2B).

For each MRI scan, datasets sampled by MRI receive coil and EMI sensing coils within the second window (i.e., EMI signals only) were utilized for training and validation. The split for the data samples were 85% for training and 15% for validation. During the training, the mean squared error (MSE) loss was minimized using Adam optimizer¹⁰ with $\beta_1 = 0.9$, $\beta_2 = 0.999$, and initial learning rate = 0.0005. The CNN model was implemented with a batch size of 16 for 20 epochs. For the testing, datasets sampled by both MRI receive coil and EMI sensing coils within the first window (i.e., MRI + EMI signals) were utilized.

- iii. We believe that other possibilities do exist to further advance deep learning EMI elimination (or raw MRI “signal denoising”) strategies in the future using various deep learning procedures. For example, the network or model could be trained using not only the data acquired during a particular scan protocol

for a specific subject, but also in combination with other patient-, scanner- or/and environment-specific data from the same subject or/and data available from other subjects. Nevertheless, these concepts are beyond the focus of this manuscript, and we hope to interrogate in the future once we gain more experience under truly clinical settings.

- iv. In this revision, we have included the above technical details regarding the implementation, training procedures, and neural network architecture in the ‘*Deep learning driven EMI detection and elimination*’ subsection of the methods section.

R1-1d It is claimed that the EMI cancellation procedure "obtained phantom and brain image SNRs that were $\geq 95\%$ of those when using a fully enclosed RF shielding cage for direct comparison." Since no comparison to an acquisition with an RF-shield was performed, this statement should be removed.

- i. We apologize to the reviewer here for not including any data to support this statement in the original submission.
- ii. We indeed performed such experiments. We directly compared the two scenarios, where a fully enclosed RF shielding cage was installed or removed, respectively, as illustrated in **Fig. R1-1d-1** below. **Fig. R1-1d-2** below show the measurements and analyses to quantify the EMI noise levels in the images.

As quantified and demonstrated in **Fig. R1-1d-2** below, our proposed deep learning driven EMI cancellation method was able to provide nearly complete removal of EMI noise in the final images. That is, in absence of EMI shielding cage, our method produced final image noise levels as low as those obtained using a fully enclosed RF shielding cage (*within 5% range*) in human brain experiments. Note that we conducted this experiment using the 3D FSE FLAIR protocol for its acquisition speed (due to its short TR, i.e., 5 mins total scan time for NEX = 2).

Figure R1-1d-1. Experimental setups for quantitative assessment of the effectiveness of our proposed EMI cancellation method in reducing/eliminating EMI related image noise. For comparison, a custom-made RF shielding cage (consisting of aluminum covers) was (A) installed to fully enclose the subject, or (B) removed.

Figure R1-1d-2. In absence of RF shielding, our proposed deep learning driven EMI cancellation method achieves brain image noise levels as low as those obtained with the installation of a fully enclosed RF shielding cage (within 5% range). Representative 3D FSE FLAIR images with NEX = 2 (left; as average of two individual images), corresponding noise images (right; as the difference between two individual images), and noise level quantifications (A) before and (B) after EMI elimination. Two imaging experiments (with and without RF shielding cage) were performed in the same normal adult subject. Frequency encoding (FE) direction was along the vertical direction. Note that, before EMI cancellation, a reduced and stable narrowband internal EMI signal was still present in the images when RF shielding cage was installed (due to MRI console EMI leakage, as indicated by green arrows). Without RF shielding cage, both narrowband and wideband external EMI signals were present (indicated by red arrows). The image noise levels were quantified here from the difference image between the first and second individual complex images¹¹. Without deep learning EMI elimination procedure, the average noise level without RF shielding cage was significantly higher (1.423 in standard deviation, SD) than that with RF shielding cage (0.581) as expected. After deep learning EMI elimination, the average noise level without RF shielding cage significantly decreased to 0.588 (from 1.423), which was only 1.2% higher than that obtained with RF shielding cage before EMI elimination (0.581) and 4.8% higher than that after EMI elimination (0.561). The EMI elimination procedure provided nearly complete removal of EMI noise in the images. That is, in absence of EMI shielding cage, it led to final image noise levels as low as those obtained with a fully enclosed RF shielding cage installed (within 5% range) in human brain experiments. No conductive shielding cloth or EMI pickup electrodes were used in this study.

iii. In this revision, we have included the results in **Fig. R1-1d-2** above as a new **Supplementary Fig. 3**. We have also clarified the relevant statements in the ‘0.055 T brain ULF MRI system hardware design’ subsection of the results section.

We have also added details regarding our experimental measurements and associated analyses to quantify the EMI noise levels in human brain imaging experiments, with and without the full RF

shielding cage in the ‘*Deep learning driven EMI detection and elimination*’ subsection of the methods section.

R1-2 The brain image results appear to have almost no contrast between gray and white matter. I understand that the results cannot be compared to what we are used to seeing from current clinical systems but given that the superior soft-tissue contrast is one of the main assets of MRI over modalities like CT, I feel this should be addressed. Is this an inherent limitation at this field strength because the T1 and T2 values are much closer together, or is it just a matter of lower contrast to noise ratio? If the reason is contrast to noise ratio, this could be tested in an experiment with a higher number of averages to boost SNR?

- i. We believe that this lack of apparent gray matter (GM) and white matter (WM) contrast is caused by (a) strong partial volume or blurring effect due to low spatial resolution, i.e., 10 mm acquisition slice thickness (though displayed at 5 mm by k-space zero-padding interpolation); (b) diminished contrast to noise ratio due to high noise level at 0.055 T; and (c) our limited focus on GM/WM contrast during our protocol optimization.

So far, our protocols parameter optimization has been mostly driven by SNR optimization while preserving certain degree of useful GM/WM contrast. In fact, our FSE T2W images do present reasonably usable GM/WM contrast (see main Fig. 4B, especially in the coronal images). For FSE FLAIR images, we have mostly prioritized the SNR acquisition efficiency optimization, though GM/WM contrast could be increased with longer echo time (TE) at cost of SNR. Similarly, EPI DWI protocol has been also optimized mostly for SNR instead of GM/WM contrast, though it could be enhanced with longer TE again at cost of SNR.

T1W images normally provide the best GM/WM contrast. Our 3D GRE T1W sequence has been optimized mostly for SNR instead of GM/WM contrast, because our clinical investigators advised that T1W GM/WM optimization was of less priority since T1W image core value was to show Gd contrast enhancement rather to differentiate GM and WM.

- ii. The relative differences between GM and WM in T1 and T2 values indeed become smaller at ULF when compared with those at high field. A recent preliminary study by the Leiden group measured the T1/T2 values at 0.05 T¹² to be 330/104 ms for GM and 272/95 ms for WM vs. the 1124/95 ms and 884/72 ms at 1.5 T¹³, and 1300/110 ms and 830/80 ms at 3 T¹⁴. Using these values and Bloch equation derived relaxation equations¹⁵, we estimate the expected intrinsic signal contrast between GM and WM at 0.05 T and 1.5 T for our specific 0.055 T GRE T1W sequence parameters. The contrast is ~10% at 0.05 T vs. ~15% at 1.5 T and ~30% at 3 T, indicating a reduced GM/WM contrast as field decreases, but not a drastically diminished one at 0.05 T. Given the relatively low image SNRs at ULF,

the eventual GM/WM contrast to noise ratio is indeed expected to be substantially lower, making the GM and WM structures less differentiable in our T1W images.

- iii. As per reviewer's suggestion, we have conducted a 3D GRE T1W imaging experiment with increased averages by 3-fold (number of averages from the original 2 to 6, i.e., image SNR increase by $\sim\sqrt{3}$). We did not notice any visually perceptible improvement in GM/WM contrast (data not shown), likely because of the dominating partial volume effect associated with 10 mm acquisition slice thickness as well as the low in-plane resolution.
- iv. We agree that future efforts should also focus on optimizing the contrasts between various tissues. For example, inversion recovery based FSE sequences can be employed to provide better GM/WM contrast as commercial company Hyperfine has done¹⁶.
- v. In this revision, we have added few short statements from the discussions above regarding contrast optimization in the '*Imaging protocol implementation using 0.055 T brain ULF MRI*' subsection of the results and '*Challenges of imaging at ultra-low-field*' subsection of the discussion section, respectively.

We have also carefully re-analyzed our preliminary raw data for GM/WM T1 estimation at 0.055 T and updated WM T1 value to ~ 260 ms. This value has been updated in the revised '*Promises of imaging at ultra-low-field*' subsection of the discussion section.

R1-3 It is stated that "Scanning was acoustically much quieter when compared to high- field MRI, with maximum peak sound pressure level (SPL) <85 dBA at 0.055 T (Supplementary Fig. 1) vs. ≤ 120 dBA at 3T." I do not doubt that the scanner creates less acoustic noise, but the statement suggests that an actual measurement of the noise level was performed at 3T as well, with matched experimental settings and pulse sequences. Unless the noise measurement was done exactly in the same way and with the same sequences as in (30), I suggest tone down this statement.

- i. We agree that the statement made here is misleading as we did not perform the SPL measurements at 3 T with matched experimental settings.
- ii. In this revision, as suggested by the reviewer, we have clarified this issue accordingly in the '*Imaging protocol implementation using 0.055 T brain ULF MRI*' subsection of the results section.

R1-4 Sequence parameters

R1-4a Please provide the echo train length of the 3D fast spin echo acquisitions

- i. The echo train length (ETL) for 3D FSE T2W protocol was 21. For 3D FSE FLAIR protocol, ETL was 13.
- ii. In this revision, we have included these ETL values in the '*ULF MRI scan protocols and optimization*' subsection of the methods section.

R1-4b Please clarify how the reconstructed image resolution of $1 \times 1 \times 5 \text{ mm}^3$ was achieved and why this step was performed. Was this done with k-space zero padding, and was the reason to make the images more comparable to 3T for the clinical study?

- i. Zero padding in k-space was applied to achieve image display resolution of $1 \times 1 \times 5 \text{ mm}^3$ resolution. This image interpolation procedure via k-space zero padding was performed for better visualization effect and comparison to high-resolution 3T images.
- ii. In this revision, we have further clarified these post-processing procedures in the '*ULF MRI scan protocols and optimization*' subsection of methods section.

R1-4c Please report sequence parameters for the standard clinical 3T protocols as well.

- i. The sequence parameters for our 3 T clinical protocols are as follows.

T1W and T2W images were both acquired with 2D FSE. T1W sequence parameters were: TR/TE = 2700/25 ms, inversion time = 830 ms, FA = 111° , ETL = 8, acquisition matrix = 340×280 , FOV = $230 \times 230 \text{ mm}^2$, acquisition slice thickness = 5 mm, 27 slices, and NEX = 1. T2W sequence parameters were: TR/TE = 5900/106 ms, FA = 120° , ETL = 30, acquisition matrix = 448×448 , FOV = $230 \times 230 \text{ mm}^2$, acquisition slice thickness/slice gap = 5/0.5 mm, 27 slices, and NEX = 2.

FLAIR images were acquired with 3D FSE with TR/TE = 6300/104 ms, inversion time = 1800 ms, FA = $90/180^\circ$, ETL = 180, acquisition matrix = $256 \times 256 \times 60$, FOV = $250 \times 250 \times 150 \text{ mm}^3$, acquisition slice thickness = 2.5 mm, and NEX = 1.

DWI images were acquired using a 2D spin-echo EPI using a diffusion gradient pair. The parameters were TR/TE = 4000/57 ms, acquisition matrix = 120×160 , FOV = $230 \times 230 \text{ mm}^2$, acquisition slice thickness = 5, 54 slices, NEX = 4 for images with $b = 0$ (i.e., b_0 images) and images with $b = 1000 \text{ s/mm}^2$ (i.e., b_1 images) diffusion weighting along three orthogonal directions.

- ii. In this revision, we have included these 3T sequence parameters above in the '*Study participants and clinical 3 T MRI scans*' subsection' of the methods section.

R1-5 Please provide more details about the clinical study. Was the radiologist fully blinded to the corresponding 3T images and the clinical presentation of the patients, or did he/she read those as well at a different point in time, or was this done by a different radiologist? Was the only criterium presence/absence of pathology? Were there cases where the diagnosis on 3T and 0.055T did not match, and in that case would that have led to a different therapeutic decision?

- i. The 0.055 T patient images were not used for making any diagnostic or/and subsequent therapeutic decisions. Only the 3T images were clinically reported by the patients' attending radiologist (not any of the clinical co-authors in this study) for subsequent clinical management.

In this study, the 0.055 T and 3 T images were read on the same day by one senior clinical radiologist (co-author H.K. Mak). The 0.055 T images were read first. As such, they were blinded to the corresponding 3 T images when evaluating the ULF images. The respective patient's clinical history was made available, which was a standard information available to any attending neuroradiologist or physician before reading and reporting the MRI findings. The primary criteria for image evaluation were to determine whether and what specific lesions could be observed from the 0.055 T images. There were cases where 0.055 T images did not correspond entirely with 3 T images (such as in main Fig. 7B). This discrepancy has been commented when describing the observation in the results section. Specifically, the 0.055 T T1W images showed weaker contrast at the hematoma (i.e., hyperintense rim and hypointense core) likely due to a combination of factors such as (1) 0.055 T biophysics, which remains to be fully investigated when visualizing blood product and/or hemosiderin of different stages at the rim and core regions, and (2) optimization of T1W contrast as discussed in **R1-2** above.

- ii. We recognize that, before the eventual clinical adoption, ULF MRI needs more detailed and comprehensive tests to determine its diagnostic sensitivity and specificity in detecting pathologies across a spectrum of neurological diseases under specific clinical settings. Presently, as the next step forward, we are in discussions with our clinical collaborators to assemble another 0.055 T ULF MRI scanner, site and operate under a truly clinical setting (such as inside the stroke center) for diagnostic purposes, and evaluate its clinical utility and efficacy.
- iii. In this revision, we have provided more details about how the images were read by our clinical co-authors and the criteria for evaluation of the ULF images in the '*Study participants and clinical 3 T MRI scans*' subsection of the methods section, as discussed above.

R1-6 Please indicate the direction of the B_0 field in Figure 8?

- i. The direction of B_0 field is along the vertical direction for all axial 3D GRE T1W and 3D FSE T2W images in Fig. 8. See **Fig. R1-6** below, where B_0 field direction is indicated.

Figure R1-6. Low sensitivity to commonly used clinical metal implants at 0.055 T. Illustration of metal implants (left) and corresponding images acquired at 0.055 T (right) of (A) titanium alloy aneurysm clips and, (B) cerebrovascular stents with three distinct types of metal alloys. Metal-induced image artifacts were dramatically reduced at 0.055 T. Almost no visible artifacts were present around the titanium alloy metal clips and the nickel-titanium alloy stent. Slight artifacts were visible in the images for the cobalt-platinum alloy and stainless-steel stents (indicated by red arrows). These implants were immersed in water and imaged with the 3D axial GRE and FSE sequences with frequency encoding (FE) direction along horizontal or vertical direction. Note that the main field 0.055 T was along the vertical direction.

ii. In this revision, main Fig. 8 has been replaced with Fig. R1-6 above.

R1-7 p11: To some degree, the section "Promises of imaging at ultra-low-field (p11)" is redundant to the introduction. This should be streamlined.

In this revision, a substantial portion of the first paragraph in the discussion subsection has been removed as it repeats the general statements that are already present in the introduction section.

However, we decided to retain the detailed discussions of the advantages of imaging at ULF (e.g., low acoustic noise levels during scanning, low sensitivity to metallic implants, and unique ULF biophysics etc.) in the hope of inspiring future and broad developments of low-cost ULF MRI technologies for improved accessibility across various healthcare sectors, and combating healthcare disparities.

R1-8 It is mentioned that at ULF, the noise in MRI signals is dominated by the RF receiver coil noise, while the sample noise is negligible. Did that influence the design of the RF coil that was built for the scanner?

- i. At ULF, the noise in proton MRI signal is dominated by the noise in the RF receive coil and subsequent front-end preamplifier electronics, in contrast to body thermal noise at high field.

This is a valuable phenomenon that holds great potential for future ULF technology development. However, it didn't influence our RF coil design in this study. As mentioned in the methods section of the manuscript, our head MRI receive coil was a classical one-channel solenoid room temperature RF coil^{17,18}, followed by a two-stage amplification using low-noise amplifiers.

At this moment, there has been no attempt to reduce RF coil noise and improve image SNR at ULF through cryogen-cooled¹⁹ or conduction-cooled (now possible as recently demonstrated by conduction-cooled cryogen-free superconducting magnet) RF coil and front-end RF electronics. In our view, with better and cheaper cryo-coolers becoming available, conduction-cooled RF coils can represent an important direction in future ULF hardware development.

- ii. In this revision, we have made clarifications regarding the RF coil design as above in the '*Gradient and radiofrequency (RF) subsystems*' subsection of the methods section.

R1-9 I assume that a total of 34 patients were recruited for the study, since 6+3 ended up not being included, and the total for the study was still 25 (page 20 lines 472 to 475).

- i. We thank the reviewer for drawing our attention to this error in the manuscript. We indeed recruited a total of 34 patients. Six patients dropped out of the study due to deteriorating medical condition before scheduled scans and a further three patients were excluded due to physical discomfort or chest access issues during 0.055 T scan. As a result, a net total of 25 patients completed the study.
- ii. In this revision, we have corrected this error in the '*Study participants and clinical 3 T MRI scans*' subsection of the methods section.

R1-10 I appreciate that data will be made available publicly and custom computer codes will be made available from the corresponding author upon request. In the spirit of reproducible research, I would appreciate it even more if those were included in the public repository as well. The strongest impact would of course be achieved if the authors also shared hardware plans and component lists, which would allow other research groups to reproduce the authors' design

- i. We agree with the reviewer's suggestion, and shall share our key designs, codes and all data presented in manuscript in a public repository.

We will share publicly the key hardware designs, software codes and data in GitHub (<https://github.com/bispmri/Ultra-low-field-MRI-Scanner>) upon manuscript publication. They include but are not limited to: (1) detailed technical design documents of our 0.055 T permanent SmCo magnet with optimal eddy current and homogeneity performance, which is the most guarded information by all commercial permanent magnet MRI developers/manufacturers; (2) codes and data to replicate our deep learning based EMI elimination method and results, as in Fig. 3 and the new Supplementary Figs. 1-3; and (3) k-space data that correspond to all image results from healthy subjects and patients shown in the manuscript.

We will also prepare and share a list of key hardware components and their vendor information in the public repository.

- ii. In this revision, we have revised the data and code availability statement to indicate the availability of all key designs, codes and data that support the findings of this study in a public repository, as well as the availability of other information upon request.

We have also revised the ‘*Magnet design*’ subsection of the methods section to include the exact material types of our 0.055 T samarium-cobalt (SmCo) magnet components, which are key information for replicating our high-performance SmCo magnet design.

We believe such sharing will not only maximize reproducibility, but also promote and expedite the further innovation of core ULF MRI technologies, especially in areas of low-cost but highly functional magnet designs, new EMI suppression and raw MRI signal “denoising” procedures/algorithms, and deep learning assisted ULF image reconstruction so to increase image quality, as discussed in the discussion section of the manuscript. Thank you for the suggestion.

Reviewer #2:

R2-G (General Comments) This paper is well-written and presents the highest quality low-field MR images that I have seen. Certainly, they are a massive improvement on the highly distorted images shown in a previous publication in Nature Communications. An impressive degree of EMI reduction is shown with the neural network, and images using highly B₀-sensitive sequences such as EPI and TrueFISP are presented for the first time.

- i. We thank the reviewer for his/her strong compliments on the ultra-low-field (ULF) imaging capability and ULF image quality reported in this study.

Indeed, we consider the implementation, optimization and demonstration of the four *essential* clinical MRI neuroimaging protocols at 0.055 T one of the key contributions by this study. Technically this is significant, because sequence such as echo-planar imaging (EPI) diffusion-weighted imaging (DWI) is extremely hardware demanding and its feasibility at ULF has not been successfully demonstrated for brain imaging by anyone to our knowledge. All ULF MRI studies from others^{6,20-22} so far only demonstrated the relatively simple gradient-echo (GRE) and fast spin-echo (FSE) sequences (and often with suboptimal image quality), except commercial company Hyperfine who implemented and demonstrated FSE based DWI but with severe image artifacts¹⁶. To our knowledge, our present study demonstrated the brain EPI and EPI based DWI at ULF for the first time, and EPI and DWI image quality were free from any substantial artifacts.

- ii. Note that our “massive improvement” of ULF image quality was made possible partly by the magnet specifications we defined five years ago at the start of our ULF research and development project. A key consideration here was to design a magnet for a reasonably high B_0 homogeneity within a sufficiently large field-of-view (FOV) that can provide proper coverage of entire human brain. Our 0.055 T ULF magnet had an inhomogeneity of <250 ppm peak-to-peak over 240 mm diameter of spherical volume (DSV). As such, the images did not generally suffer from severe gross geometric distortions.

As the reviewer insightfully stated, sequences such as EPI DWI are highly vulnerable to B_0 issues including eddy currents. As shown in main Fig. 1B, our ULF magnet incorporated an anti-eddy current plate design to mitigate the eddy current problems. The high EPI DWI quality was also made possible partly through our technical experience in dealing with various EPI artifact problems in the past ten years^{11,23-25}.

- iii. We did notice the recent paper from Yale University that appeared in Nature Communications²⁶ on 8/25/2021 after our original manuscript submission to Nature Communications on 7/22/2021. It focused on demonstrating the clinical value of commercial and proprietary 0.064 T Hyperfine head scanner in evaluating intracerebral hemorrhage. The paper didn't reveal any key information regarding critical scanner design and imaging protocol details. As the reviewer commented, its image quality was suboptimal.
- iv. *We hope and expect that the ULF imaging capability and image quality reported in this manuscript, together with the methods and in-depth technical details disseminated in this revision, will motivate other MRI researchers to focus on innovating imaging methods on low-cost ULF MRI platforms, so to tackle the greater issue of MRI accessibility within healthcare in both developed and underdeveloped countries.*

R2-1 In terms of science the main novelty is the means of EMI reduction. Certainly, the ability to operate such a system in an RF noisy environment is critical, but several other groups have shown that very simple shielding can probably suffice, so it is unclear whether this is a true breakthrough. For example, O'Reilly et al. have shown that a simple conductive cloth eliminates the vast majority of EMI, and the Hyperfine unit also incorporates a (commercially proprietary) highly effective EMI reduction scheme.

- i. We thank the reviewer for this comment and sharing his/her knowledge of this field.

We believe the contributions of our work go beyond the EMI elimination method. We consider that our work represents a significant research and engineering milestone in development of a new class of technologies to enable patient-centric and site-agnostic MRI scanners to fulfill the unmet clinical needs across various healthcare communities and for the developing and underdeveloped world. Specifically, we demonstrated critical and vital components for the clinical utility of such low-cost scanners with our (1) SmCo permanent magnet design choices, (2) an active deep learning driven EMI detection, prediction and cancellation strategy for RF shielding-free deployment, (3) essential clinical neuroimaging protocols, and (4) initial clinical study comparing our ULF tumor and stroke patient images with those acquired at clinical high-field scanners.

The significance of these contributions is also strongly echoed by Reviewer #1's general comments in **R1-G** above.

- ii. As stated in **R1-G-3** above and in the paragraphs below, *we consider our deep learning driven active EMI cancellation strategy novel and highly effective.* This claim is supported and demonstrated by the results presented in the original submission, and further supported by the additional results and analyses presented in this rebuttal and revision (see **R1-1a** to **R1-1d** above for details).

We started tackling this active EMI elimination challenge for low-cost and radiofrequency (RF) shielding-free MRI five years ago by first defining the key requirements as follows. Our key considerations for EMI removal strategy are as follows: (a) The developed method must achieve effective and nearly total EMI removal when compared to the ground truth scenario (, i.e., when an RF shielding cage is deployed to fully enclose the subject during MRI scan). (b) The method must be able to deal with EMI signals that changes dynamically over time during MRI scan. In reality, such changes can arise from the surrounding EMI sources with various nature and behaviors. EMI signal received by MRI receive coil can be also influenced by the human body, which serves as an effective antenna^{1,2} for EMI reception in a shielding-free MRI setting. For example, varying body size and weight can alter the level and characteristics of EMI signal picked up by body and subsequently detected by MRI receive coil. Human body position change during MRI scan can also alter the EMI signal detected by MRI receive coil due to the change in electromagnetic coupling between surrounding EMI emitting sources and receiving human body. (c) The EMI signal from scanner

internal low-cost MRI electronics (e.g., gradient and RF amplifiers, console, power supplies, and their suboptimal insulations) must be dealt with as well. *Therefore, an active, accurate, highly resilient and relatively simple EMI removal strategy is highly desired for RF shielding-free and low-cost ULF MRI. Such successful strategy is a mandate if one wishes to implement the intrinsic low-SNR MRI protocols at ULF such as diffusion-weighted imaging, a key clinical neuroimaging protocol.*

We subsequently solved this active EMI detection and cancellation problem for RF shielding-free MRI by **(a)** taking advantage of the well-established multi-receiver MRI electronics (previously developed for parallel imaging) and **(b)** utilizing separate resonant or tuned EMI sensing RF coils to simultaneously acquire EMI signals in various spatial locations during two windows, namely, MRI signal acquisition and EMI signal characterization windows (as illustrated in main Fig. 2). We derive and establish a model to predict the EMI signal in MRI receive coil from the EMI signals detected by EMI sensing coils. Intuitively, an accurate prediction model should be highly feasible because of a simple electromagnetic phenomenon. That is, the properties of RF signal propagations among any radiative (e.g., air) or/and conductive media (e.g., surrounding EMI emitting structures such as power lines and nearby equipment, RF coils, other MRI hardware pieces and cables, patient bed, imaging object or human body) are fully dictated by the electromagnetic coupling among these media or structures. Such coupling relationships can be analytically characterized in a relatively simple manner by the frequency domain coupling or transfer functions among structures (e.g., among MRI receive coil and sensing coils).

With the considerations above, we opted to develop and deploy a deep learning driven approach to derive this model for establishing the relationships among the EMI signals detected by EMI sensing coils and MRI receive coil. *We hypothesized that the nature of deep learning should provide an accurate and robust EMI prediction and cancellation procedure, yet relatively simple for shielding-free MRI in various unshielded imaging environments. Indeed, this belief is well supported by our results in both original submission and the revised submission here.*

Specifically, in this revision submission, we further demonstrate that our deep learning driven EMI elimination strategy worked robustly with dynamically varying EMI from both external environments and internal electronics (see new **Supplementary Figs. 1 and 2** and **R1-1a & 1d** above in response to Reviewer #1's comments), even in presence of subject body motion during scan (see **R1-1a-iv** above in response to Reviewer #1's comments). *More importantly*, we also compared our EMI cancellation scheme performance directly to the ground truth scenario, i.e., when a fully enclosed RF shielding cage was installed to cover the subject (see new **Supplementary Fig. 3** and **R1-1d** above in response to Reviewer #1's comments). *These results directly demonstrated the highly robust and effective EMI removal through our proposed strategy, producing final image noise levels as low as those obtained using a fully enclosed RF shielding cage (within 5% range) in human brain experiments. Note that no*

*conductive cloth or any body EMI pickup electrodes (as mandated in the Vanderbilt/MGH approach as discussed below in **R2-1-iii**) were used in our study.*

Last, we believe our deep learning driven EMI elimination strategy will benefit MRI in general, including high field MRI. We are currently exploring the use of our proposed strategy to remove EMI related artifacts/noise in a 1.5 T scanner (arising from hardware imperfections or/and opening of RF shielding room door). Preliminary results are positive but they are beyond the scope and focus of the current manuscript. Our proposed deep learning EMI removal strategy is also novel to EMI-sensitive RF signal detection devices and equipment industry to our best knowledge. We believe that our proposed simultaneous and data-driven learning of the complex EMI environments and operation scenarios will likely present a powerful approach to accurately map and eliminate EMI noise in other non-MRI RF signal detection applications.

- iii. We're also aware that three other academic groups have been actively seeking solutions to mitigate EMI issues for imaging in unshielded environments. In 2018, one group proposed to use 3 orthogonal magnetometers and one 2nd order gradiometer to sense environmental EMI and then estimate the EMI signal in MRI receive coil for ULF through an adaptive estimation and suppression procedure for ULF imaging³. However, the results were suboptimal and the approach was hardware demanding.

Group from Leiden have attempted to use simple conductive cloth to cover the subject during scan⁶. This passive method can alter and reduce EMI signal mainly from external environments. *However*, its performance is far from optimal. It is also inadequate to deal with potential EMI sources from scanner internal electronics.

Group from Vanderbilt/MGH are presently working on analytical approaches to estimate and remove external EMI signal received by MRI receive coil using tuned EMI sensing RF coils (as reported in their preliminary studies during August 2020 ISMRM Annual Meeting⁴ and May 2021 ISMRM Annual Meeting⁷). Their first approach was based on the frequency domain transfer function concept as discussed in **R1-G-3-iii** and implemented in frequency domain but with only limited preliminary results for demonstration⁴. The second approach was also based on the frequency domain transfer function concept but implemented in the time domain as linear convolutions under the *assumptions* that their corresponding transfer functions (between MRI receive coil and EMI sensing coils in frequency domain) were relatively smooth within MRI signal receiving bandwidth *and* stable during scan (so finite and small convolution windows could be used for linear convolution operations, partly resembling the GRAPPA parallel imaging concept⁸), and EMI spectra should be generally bandlimited. This approach analytically modeled the relationship between EMI signals simultaneously detected by MRI receive coil and EMI sensing coils through time domain linear convolutions based on these specific assumptions, thus can be (a) potentially vulnerable to changes in the transfer

functions during scan (e.g., due to patient body position change or movement of nearby attending staff or equipment as discussed in **R1-G-3-iii** above), **(b)** potentially problematic given that EMI spectra can be often broadband instead of bandlimited or compact within the MRI sampling bandwidth. The Vanderbilt/MGH group partly mitigated these issues (mostly issue a) by analyzing and subdividing the dataset to sub-datasets to accommodate these EMI signal characteristic changes during scan. *However*, such adaptive procedure, given the assumptions regarding transfer functions (vs. the reality as discussed above), likely would limit the robustness of their proposed EMI cancellation strategy. Their preliminary results demonstrated substantial EMI removal *but required the use of electrodes mounted on body surface as well as conductive cloth to cover subject*⁶ in order to achieve highly effective EMI removal, which we consider also cumbersome in practice. *Further*, the efficacy of this latter strategy also remains to be fully assessed through a cohort of subjects or/and patients (over 100 subjects/patients in our study so far instead of few subjects only in their study) in terms of removing both external and internal EMI without introducing artifacts. Note that we aimed to target and eliminate both external EMI and EMI from scanner internal electronics in our present study.

- iv. As pointed by the reviewer, commercial company Hyperfine has recently demonstrated a 0.064 T head MRI scanner without RF shielding room requirement (www.hyperfine.io/portable-mri) using an undisclosed and proprietary EMI removal method. Although its details (as well as other key information regarding scanner hardware and imaging sequences) are unavailable, we speculate that it is based on the use of frequency domain transfer function approach and EMI sensing coils⁵ yet we cannot confirm it with certainty.
- v. In this revision, as also suggested by Reviewer #1, we have cited more literature, added statements and references regarding the present landscape of EMI strategies and why we chose to pursue a deep learning based approach, as discussed above (as well as in **R1-G-3** and **R1-1** above).

Further, new experiments and analyses have been performed and added regarding the robustness and effectiveness of our deep learning driven EMI elimination strategy (new **Supplementary Figs. 1 to 3**).

R2-2 There are a few references that should be added, for example the issue of reduced artifacts from metallic implants has been covered by van Speybroeck et al in 2021.

We thank the reviewer for alluding us to this recent study from van Speybroeck et. al²⁷ on the examination of metal implants at several low-field strengths (<0.1 T).

In this revision, we have acknowledged this particular work when describing the advantages of imaging at ULF such as less mechanical forces and RF-induced heating, and absence of gross metal-induced artifacts in the *'Promises imaging at ultra-low-field'* subsection in the discussion section.

We have also cited a recent study²⁸ that documented improved safety and reported absence of gross metal-induced artifacts when imaging various regions of interest at 0.2 T for patients with cardiac rhythm management devices (e.g., pacemakers).

The authors would like to take this opportunity to thank the two reviewers for their valuable critiques and suggestions, which we believe have substantially strengthened this manuscript.

References

1. Kibret, B., Teshome, A.K. & Lai, D.T.H. Analysis of the Human Body as an Antenna for Wireless Implant Communication. *Ieee Transactions on Antennas and Propagation* **64**, 1466-1476 (2016).
2. Sen, S., Maity, S. & Das, D. The body is the network: To safeguard sensitive data, turn flesh and tissue into a secure wireless channel. *Ieee Spectrum* **57**, 44-49 (2020).
3. Huang, X., Dong, H., Qiu, Y., Li, B., Tao, Q., Zhang, Y., Krause, H.J., Offenhausser, A. & Xie, X. Adaptive suppression of power line interference in ultra-low field magnetic resonance imaging in an unshielded environment. *J Magn Reson* **286**, 52-59 (2018).
4. Srivinas, S.A., Cooley, C.Z., Stockmann, J.P., McDaniel, P.C. & Wald, L.L. Retrospective electromagnetic interference mitigation in a portable low field MRI system. in *Proceedings of International Society of Magnetic Resonance in Medicine* p1269 (August 2020).
5. Rearick, T., Charvat, G.L., Rosen, M.S. & Rothberg, J.M. Noise suppression methods and apparatus. (US Patent No. 9,797,971). U.S. Patent and Trademark Office (2017).
6. O'Reilly, T., Teeuwisse, W.M., de Gans, D., Koolstra, K. & Webb, A.G. In vivo 3D brain and extremity MRI at 50 mT using a permanent magnet Halbach array. *Magn Reson Med* **85**, 495-505 (2021).
7. Srivinas, S.A., Cauley, S., Stockmann, J.P., Sappo, C.R., Vaughn, C.E., Wald, L.L., Grissom, W.A. & Cooley, C.Z. In vivo human imaging on a 47.5mT open MRI system with active Electromagnetic Interference (EMI) mitigation using an electrode. in *Proceedings of International Society of Magnetic Resonance in Medicine* p4030 (May 2021).
8. Griswold, M.A., Jakob, P.M., Heidemann, R.M., Nittka, M., Jellus, V., Wang, J., Kiefer, B. & Haase, A. Generalized autocalibrating partially parallel acquisitions (GRAPPA). *Magn Reson Med* **47**, 1202-1210 (2002).
9. Souza, S.P., Szumowski, J., Dumoulin, C.L., Plewes, D.P. & Glover, G. SIMA: simultaneous multislice acquisition of MR images by Hadamard-encoded excitation. *J Comput Assist Tomogr* **12**, 1026-1030 (1988).
10. Kingma, D.P. & Ba, J. Adam: A method for stochastic optimization. *arXiv preprint arXiv:1412.6980* (2014).
11. Xie, V.B., Lyu, M. & Wu, E.X. EPI Nyquist ghost and geometric distortion correction by two-frame phase labeling. *Magn Reson Med* **77**, 1749-1761 (2017).
12. O'Reilly, T. & Webb, A.G. High resolution in-vivo relaxation time mapping at 50 mT. in *Proceedings of International Society of Magnetic Resonance in Medicine* p3084 (May 2021).
13. Stanisz, G.J., Odobina, E.E., Pun, J., Escaravage, M., Graham, S.J., Bronskill, M.J. & Henkelman, R.M. T1, T2 relaxation and magnetization transfer in tissue at 3T. *Magn Reson Med* **54**, 507-512 (2005).
14. Wansapura, J.P., Holland, S.K., Dunn, R.S. & Ball, W.S. NMR relaxation times in the human brain at 3.0 tesla. *J Magn Reson Imaging* **9**, 531-538 (1999).
15. Magnetization, Relaxation, and the Bloch Equation. in *Magnetic Resonance Imaging* (eds. Brown, R.W., Cheng, Y.N., Haacke, E.M., Thompson, M.R. & Venkatesan, R.) 53-66 (2014).
16. Sheth, K.N., *et al.* Assessment of Brain Injury Using Portable, Low-Field Magnetic Resonance Imaging at the Bedside of Critically Ill Patients. *JAMA Neurol* **78**, 41-47 (2020).
17. Coffey, A.M., Truong, M.L. & Chekmenev, E.Y. Low-field MRI can be more sensitive than high-field MRI. *J Magn Reson* **237**, 169-174 (2013).
18. Gruber, B., Froeling, M., Leiner, T. & Klomp, D.W.J. RF coils: A practical guide for nonphysicists. *J Magn Reson Imaging* **48**, 590-604 (2018).
19. Resmer, F., Seton, H.C. & Hutchison, J.M. Cryogenic receive coil and low noise preamplifier for MRI at 0.01T. *J Magn Reson* **203**, 57-65 (2010).

20. Cooley, C.Z., McDaniel, P.C., Stockmann, J.P., Srinivas, S.A., Cauley, S.F., Sliwiak, M., Sappo, C.R., Vaughn, C.F., Guerin, B., Rosen, M.S., Lev, M.H. & Wald, L.L. A portable scanner for magnetic resonance imaging of the brain. *Nat Biomed Eng* **5**, 229-239 (2021).
21. He, Y., He, W., Tan, L., Chen, F., Meng, F., Feng, H. & Xu, Z. Use of 2.1 MHz MRI scanner for brain imaging and its preliminary results in stroke. *J Magn Reson* **319**, 106829 (2020).
22. O'Reilly, T., Teeuwisse, W.M. & Webb, A.G. Three-dimensional MRI in a homogenous 27cm diameter bore Halbach array magnet. *J Magn Reson* **307**, 106578 (2019).
23. Lyu, M., Barth, M., Xie, V.B., Liu, Y., Ma, X., Feng, Y. & Wu, E.X. Robust SENSE reconstruction of simultaneous multislice EPI with low-rank enhanced coil sensitivity calibration and slice-dependent 2D Nyquist ghost correction. *Magn Reson Med* **80**, 1376-1390 (2018).
24. Xie, V.B., Lyu, M., Liu, Y., Feng, Y. & Wu, E.X. Robust EPI Nyquist ghost removal by incorporating phase error correction with sensitivity encoding (PEC-SENSE). *Magn Reson Med* **79**, 943-951 (2018).
25. Liu, Y., Lyu, M., Barth, M., Yi, Z., Leong, A.T.L., Chen, F., Feng, Y. & Wu, E.X. PEC-GRAPPA reconstruction of simultaneous multislice EPI with slice-dependent 2D Nyquist ghost correction. *Magn Reson Med* **81**, 1924-1934 (2019).
26. Mazurek, M.H., *et al.* Portable, bedside, low-field magnetic resonance imaging for evaluation of intracerebral hemorrhage. *Nat Commun* **12**, 5119 (2021).
27. Van Speybroeck, C.D.E., O'Reilly, T., Teeuwisse, W., Arnold, P.M. & Webb, A.G. Characterization of displacement forces and image artifacts in the presence of passive medical implants in low-field (<100 mT) permanent magnet-based MRI systems, and comparisons with clinical MRI systems. *Phys Med* **84**, 116-124 (2021).
28. Schukro, C. & Puchner, S.B. Safety and efficiency of low-field magnetic resonance imaging in patients with cardiac rhythm management devices. *Eur J Radiol* **118**, 96-100 (2019).

REVIEWERS' COMMENTS

Reviewer #1 (Remarks to the Author):

Review for the revised version of NCOMMS-21-28454: "A Low-cost and Shielding-free Brain MRI Scanner for Accessible Healthcare"

In line with Nature Communications transparent peer review, I agree to these reviewer comments being published along with the paper.

Signed: Florian Knoll, September 28th 2021, Erlangen, Germany

The authors have performed a most comprehensive revision that addressed all my questions, and the new material has improved the manuscript even further. I have no more comments and recommend to accept the manuscript at this stage. Congratulations on your great work.

Reviewer #2 (Remarks to the Author):

The authors have done an excellent job responding to my comments on their initial submission. I have a few minor additional requests for information, see below.

1. On line 106 the authors state that the use of conductive cloth is "sub-optimal". Was this the conclusion of the original paper, or is this the current authors' interpretation?
2. Line 213: please make it clear that reconstructed data size is only for visualization purposes (one can reconstruct to any given size and report it), and that the actual spatial resolution is quite coarse compared to other studies.
3. References 24 and 55 have now been published as full papers in Magnetic Resonance in Medicine and so should be updated accordingly
4. There are 10 sensing coils reported: schematically in the supplementary material there are only six shown. Can the authors: (a) report on the size of each sensing coil (I presume they are tuned to the Larmor frequency, otherwise please state this, and (b) how many of each were positioned in which locations.
5. Can the authors briefly report on the RF coil that they used. They mention a solenoid geometry, but please give length, diameter, number of turns, conductor geometry and Q factor (loaded with the human head and unloaded).

RESPONSES AND REVISION OF NC MANUSCRIPT # NCOMMS-21-28454

A Low-cost and Shielding-free Brain MRI Scanner for Accessible Healthcare

Reviewer #1:

R1-G (Remarks) The authors have performed a most comprehensive revision that addressed all my questions, and the new material has improved the manuscript even further. I have no more comments and recommend to accept the manuscript at this stage. Congratulations on your great work

We sincerely thank the reviewer for his time and strong enthusiasm for our work.

Reviewer #2:

R2-G (Remarks) The authors have done an excellent job responding to my comments on their initial submission. I have a few minor additional requests for information, see below.

We thank the reviewer for his/her compliment and the comments/suggestions to improve the manuscript. Please see our detailed responses to his/her additional inquiries below.

R2-1 On line 106 the authors state that the use of conductive cloth is “sub-optimal”. Was this the conclusion of the original paper, or is this the current authors’ interpretation?

- i. This statement was based on the conclusion of the original paper from the group from Leiden¹.

The authors in the paper did acknowledge that SNR performance of their imaging setup with simple conductive cloth can be further improved if EMI signals from scanner internal electronics (such as the gradient amplifier) can be better mitigated or shielded.

- ii. In this revision, we decided not to make any further edits to the statement in line 106 when describing and discussing the work of the group from Leiden because the authors already acknowledge in their paper that SNR performance can be further improved.

R2-2 Line 213: please make it clear that reconstructed data size is only for visualization purposes (one can reconstruct to any given size and report it), and that the actual spatial resolution is quite coarse compared to other studies.

We thank the reviewer for the comment.

In this revision, we have clarified that reconstructed image resolution of 0.055 T images at 1x1x5 mm³ for all scan protocols were done for the purpose of better visualization effect.

R2-3 References 24 and 55 have now been published as full papers in Magnetic Resonance in Medicine and so should be updated accordingly.

Results from the two conference presentations (references 25 and 55) have been recently published by the same authors as full journal papers during the review period of our manuscript.

Kindly note that “reference 24” should be reference 25, which refers to use of time-domain implementation of transfer function approach and electrodes proposed by Vanderbilt/MGH group.

As per reviewer’s suggestion, we updated these two conference paper references (references 25 and 55 in the last revision) by replacing them with the two full journal papers in this revision.

R2-4 There are 10 sensing coils reported: schematically in the supplementary material there are only six shown. Can the authors: (a) report on the size of each sensing coil (I presume they are tuned to the Larmor frequency, otherwise please state this, and (b) how many of each were positioned in which locations.

- i. Figure 6A is mainly for illustration purpose, where we mainly show the locations of EMI sensing coils (i.e., near head holder, under patient bed, and inside electronic cabinet) and we didn’t draw all 10 sensing coils due to the limited space.
- ii. All 10 EMI sensing coils were built using copper wires and a simple solenoid design. They had a diameter of 50 mm and were tuned to the scanner Larmor frequency (2.32 MHz). Three of them were placed in the vicinity of the patient head holder. Four were placed underneath of the patient bed on patient left and right side. Three were placed inside the electronic cabinet near the console, RF amplifier, and gradient amplifier, respectively.
- iii. In this revision, we included additional information regarding the dimension and resonant frequency of the EMI sensing coils, including their respective locations in the ‘*Gradient and radiofrequency (RF) subsystems*’ subsection of the methods section.

R2-5 Can the authors briefly report on the RF coil that they used. They mention a solenoid geometry, but please give length, diameter, number of turns, conductor geometry and Q factor (loaded with the human head and unloaded).

- i. Kindly note that, as already stated in the '*Gradient and radiofrequency (RF) subsystems*' subsection of the methods section of the original manuscript, the RF receive coil was a one-channel room temperature solenoid coil with an ellipse cross-section (vertical axis 23.0 cm and horizontal axis 19.0 cm) and a Q factor of ~30 when loaded with a human head.

This solenoid coil had 10 turns of windings, 9.5 cm length, Q factor of ~31 when unloaded.

- ii. In this revision, we provided additional information on the number of turns, coil length and unloaded Q factor as above in the '*Gradient and radiofrequency (RF) subsystems*' subsection of the methods section.

The authors would like to take this opportunity to thank the two reviewers for their valuable critiques and suggestions during the review of our original submission and subsequent revision, which we believe have substantially strengthened this manuscript.

Reference

1. O'Reilly, T., Teeuwisse, W.M., de Gans, D., Koolstra, K. & Webb, A.G. In vivo 3D brain and extremity MRI at 50 mT using a permanent magnet Halbach array. *Magn Reson Med* **85**, 495-505 (2021).